# Entropy-Regularized Process Reward Model

**Hanning Zhang**♠*     **Pengcheng Wang**♡*     **Shizhe Diao**♣     **Yong Lin**◇     **Rui Pan**♠

**Hanze Dong**△     **Dylan Zhang**♠     **Pavlo Molchanov**♣     **Tong Zhang**♠

♠*University of Illinois Urbana-Champaign*     ♡*University of Toronto*
♣*NVIDIA*     ◇*Princeton University*     △*Salesforce Research*

Reviewed on OpenReview: `https://openreview.net/forum?id=cSxDH7N3x9`

## Abstract

Large language models (LLMs) have shown promise in performing complex multi-step reasoning, yet they continue to struggle with mathematical reasoning, often making systematic errors. A promising solution is reinforcement learning (RL) guided by reward models, particularly those focusing on process rewards, which score each intermediate step rather than solely evaluating the final outcome. This approach is more effective at guiding policy models towards correct reasoning trajectories. In this work, we propose an entropy-regularized process reward model (ER-PRM) that integrates KL-regularized Markov Decision Processes (MDP) to balance policy optimization with the need to prevent the policy from shifting too far from its initial distribution. We derive a novel reward construction method based on the theoretical results. Our theoretical analysis shows that we could derive the optimal reward model from the initial policy sampling. Our empirical experiments on the MATH and GSM8K benchmarks demonstrate that ER-PRM consistently outperforms existing process reward models, achieving 1% improvement on GSM8K and 2-3% improvement on MATH under best-of-N evaluation, and more than 1% improvement under RLHF. These results highlight the efficacy of entropy-regularization in enhancing LLMs' reasoning capabilities.

## 1 Introduction

Large language models (LLMs) have demonstrated remarkable performance across numerous tasks, particularly in the challenging area of mathematical reasoning (OpenAI et al., 2024; Yang et al., 2024; Dubey et al., 2024). A key factor behind these successes is the use of synthetic data, as explored in works such as Meta-MathQA (Yu et al., 2023), MAmmoTH (Yue et al., 2023), and Open-MathInstruct (Toshniwal et al., 2024b). Open-source models fine-tuned on these synthetic datasets have shown significant improvements in test accuracy on standard mathematical reasoning benchmarks such as MATH (Hendrycks et al., 2021) and GSM8K (Cobbe et al., 2021). Following supervised fine-tuning, reinforcement learning (RL) has received considerable attention as a technique for further improving models' reasoning abilities by aligning the model's behavior with human values and goals, especially after the release of OpenAI-o1 model[1].

The RL methods used in the current literature on LLM reasoning can be broadly categorized into two approaches. The first category focuses on training-time alignment, where RL is used to fine-tune the model to maximize an RL metric evaluated by a reward model. Examples include PPO (Schulman et al., 2017), GRPO (Shao et al., 2024). More recently, scaling inference-time computation has also been shown to enhance model performance by using reward models to guide LLM decoding. Techniques such as best-of-n sampling (Dong et al., 2023) and Monte Carlo Tree Search (MCTS) (Xie et al., 2024) are representative

---

*The first two authors contribute equally. Email: `{hanning5, ruip4, shizhuo2, tozhang}@illinois.edu`, `pcheng.wang@mail.utoronto.ca`, `{sdiao, pmolchanov}@nvidia.com`, `yl7690@princeton.edu`, `hanze.dong@salesforce.com`

[1]`https://openai.com/index/introducing-openai-o1-preview/`

examples of this approach. All these methods require an external reward model to provide a reward signal to update the model parameters or rank the candidate responses. Regardless of the specific RL method, the quality of the reward function remains the most crucial factor, as it sets the upper bound for the algorithm's performance.

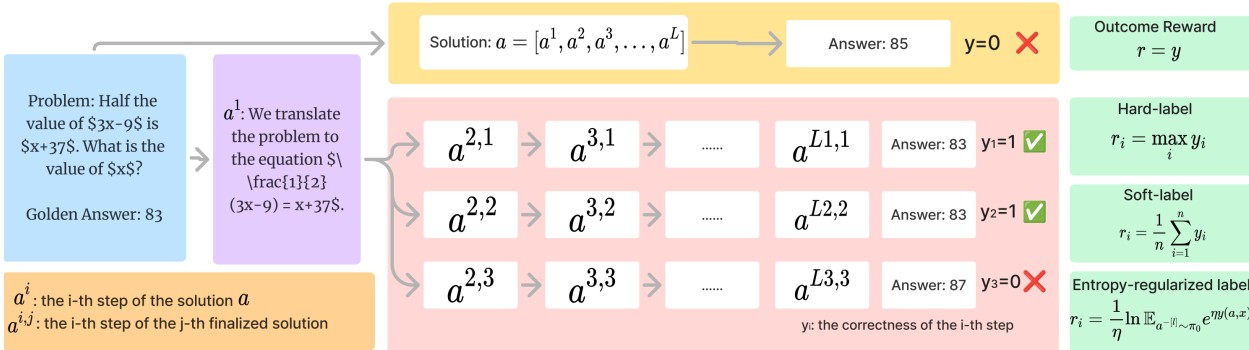

Figure 1: Illustration of ER-PRM, along with other baseline methods to construct the process reward data and outcome reward data. The key idea of ER-PRM is to calculate the process reward from the sampling trajectories under entropy-regularization.

One distinct feature of complex mathematical reasoning is that models typically require multiple steps of reasoning before producing the final answer (Wei et al., 2023). Accordingly, there are generally two types of reward models in the mathematical domain: the outcome reward model (ORM) and the process reward model (PRM). ORM provides an overall score for the quality of the final answer, while PRM assigns a score to each step in the reasoning process, providing more fine-grained feedback. Existing works (Wang et al., 2024; Lightman et al., 2023) show that PRM outperforms ORM by a considerable margin. For instance, in Lightman et al. (2023), if we use the PRM to select the final output from the candidate responses, we can achieve a test accuracy of 78.2%, as compared to the ORM with 72.4% and majority voting of 69.6%. Furthermore, PRM is more interpretable because it mirrors human behavior when evaluating reasoning paths. The PRM model is able to guide the policy model's decoding at each individual step or rerank candidate answers to enhance accuracy and consistency during generation.

Inspired by the superior performance of PRM, researchers have spent great efforts recently in the training of PRM. To get the process annotation for the PRM, one can ask humans to label the data at each step (Lightman et al., 2023), which is extremely expensive and inefficient. Since then, researchers have explored alternative labeling techniques. One representative example is the automatic labeling technique introduced by Math-Shepherd (Wang et al., 2024). The main idea of Math-Shepherd is to interpret the score of each step as its potential to deduce the correct final answer. Then, we can sample multiple trajectories starting from the intermediate step, and use the ratio of correct trajectories as a proxy for this score. The Math-Shepherd has proved as an effective way to generate high-quality process reward scores and even outperform human-labeled data.

Despite its effectiveness, Math-Shepherd considers a traditional Markov Decision Process (MDP) framework, which differs from the typical RL practice used with LLMs. Since the introduction of reinforcement learning from human feedback (RLHF), the standard approach has been using an entropy-regularized reward relative to a reference model (often the initial model) (Ouyang et al., 2022; Bai et al., 2022). This also applies to RL in reasoning tasks where we formulate the problem as an entropy-regularized MDP (Shao et al., 2024; Xiong et al., 2024b; Zhong et al., 2024b). The main goal here is to balance optimizing the reward and staying close to the original policy, which is crucial because we start with a well-trained LLM, and we do not want the model to deviate too far from this initial strong foundation policy. In recognition of this gap, in this paper, we formulate the multi-step mathematical reasoning task under the entropy-regularized MDP framework,

derive the mathematical principles of the process reward construction, and propose practical algorithms. Our contributions are summarized as follows.

- We propose the entropy-regularized process reward, a novel method for labeling the process reward score with entropy regularization.
- Based on the labeled data, we train a new process reward model to improve the performance.
- On the MATH and GSM8K datasets, we achieve significant improvements using the best-of-N evaluation method. Furthermore, by employing the reward model with rejection sampling, we observe a substantial enhancement in the performance of the policy model.
- To further boost the process reward model research, we will release all the data, code, and checkpoints to the community.

## 2  Related Work

**Mathematical Reasoning and Synthetic Data Generation.**  Mathematical problem solving is crucial for evaluating LLMs' reasoning abilities. A common approach is Chain-of-Thought (CoT) prompting, which guides LLMs to solve problems step-by-step (Wei et al., 2023; Zhou et al., 2022; Zhu et al., 2022; Tong et al., 2024). However, general chatbots still struggle with mathematical reasoning, as shown by benchmarks (Cobbe et al., 2021; Hendrycks et al., 2021). To improve performance, several works propose generating synthetic data and fine-tuning LLMs, including Meta-MathQA (Yu et al., 2023), MAmmoTH (Yue et al., 2023), MMIQC (Liu & Yao, 2024), ToRA (Gou et al., 2023), Open-MathInstruct (Toshniwal et al., 2024b), and Dart-Math (Tong et al., 2024). These datasets use teacher models to generate and filter responses, ensuring only correct answers are retained, a process known as rejection sampling or best-of-n sampling (Dong et al., 2023; Yuan et al., 2023). All these emphasize stepwise reasoning for improved task performance.

**Applying RL to Mathematical Reasoning.**  Inspired by the success of RL in chatbots like Chat-GPT and Claude, researchers have begun applying RL to mathematical reasoning tasks. Some works use deep RL methods like PPO (Schulman et al., 2017) and GRPO (Shao et al., 2024) to optimize external reward signals. Others apply direct preference learning algorithms, such as DPO and KTO (Jiao et al., 2024; Yuan et al., 2024), for problem-solving. Online iterative DPO, initially for chat tasks (Xiong et al., 2024a; Xu et al., 2023), has been adapted for CoT reasoning (Xie et al., 2024; Xiong et al., 2024b; Pang et al., 2024). Additionally, inference-time RL methods like best-of-n sampling (Dong et al., 2023; Lightman et al., 2023) and MCTS (Xie et al., 2024; Chen et al., 2024a; Lai et al., 2024) guide decoding or response selection using external reward models. These approaches, including deep RL, iterative preference learning, and inference-time methods, all depend on reward models and can benefit from our study.

**Reward Modeling in Mathematical Reasoning.**  In mathematical reasoning, reward models in RLHF are not trained via pairwise comparisons due to the availability of ground-truth answers (Bai et al., 2022; Ouyang et al., 2022; Dong et al., 2024). These models are generally divided into Outcome Reward Models (ORM) and Process Supervision Reward Models (PRM). ORM evaluates the overall quality of a response, while PRM provides feedback at each step of the reasoning process. The success of PRM approaches, like Lightman et al. (2023), has fueled interest in studying PRM construction. However, PRMs require extensive human annotation, which is challenging for open-source efforts. To mitigate this, Math-Shepherd (Wang et al., 2024) proposed a method for efficiently acquiring process reward labels by generating parallel completions and using trajectory ratios to evaluate correctness. This has been proved as effective as human annotation to generate high-quality datasets. Building on this, studies have incorporated process supervision and MCTS to improve RLHF pipelines, such as stepwise DPO with MCTS (Chen et al., 2024b), an MCTS-based PRM framework (Luo et al., 2024), MCTS Self-Refine (Zhang et al., 2024), and process generation from final answers (Chen et al., 2024a). These efforts have significantly enhanced process supervision efficiency, motivating the use of MCTS for high-quality data generation.

## 3 Approach

In this section, we present a novel reward formulation from an entropy perspective that utilizes KL-Regularization to derive the reward calculation of PRM, which we refer to as Entropy-Regularized PRM (ER-PRM). This formulation is followed by the derivation of obtaining the optimal policy model through RLHF using our entropy-regulated process reward calculation approach. The mathematical result is essentially from the KL-regularized RL literature, which has recently been studied in the context of DPO training in RLHF (Xiong et al., 2024b; Zhong et al., 2024a). And we move a step forward for the reward modeling. Subsequently, we compare our approach with Wang et al. (2024); Luo et al. (2024) and then demonstrate that our methodology is more robust than those proposed aggregation-based reward calculation due to their dependency on initial policy model.

### 3.1 Task Formulation and Optimal Policy

We consider training an LLM for reasoning. Given prompt $x$, the LLM produces $L$-step reasoning chain $a = [a^1, \ldots, a^L]$. The reward $r(a, x)$ indicates whether the result of the reasoning chain is correct or not. We want to find LLM, denoted by a policy $\pi_*(a|x)$, that optimizes $\pi \in \Pi$ with the KL-regularized loss function

$$\mathcal{L}(\pi) = -\mathbb{E}_x \mathbb{E}_{a \sim \pi(\cdot|x)} \left[ r(a, x) - \frac{1}{\eta} \ln \frac{\pi(a|x)}{\pi_0(a|x)} \right]$$

where $\pi_0$ is the initial policy model, a pretrained LLM, and $\pi$ is the model being fine-tuned. According to Lemma 1, the minimizer for $\mathcal{L}$ is

$$\pi_*(a|x) = \frac{\pi_0(a|x) e^{\eta r(a,x)}}{\mathbb{E}_{a \sim \pi_0} e^{\eta r(a,x)}} \propto \pi_0(a|x) e^{\eta r(a,x)}.$$

### 3.2 Multi-step Reasoning

We refer to multi-step reasoning as over partial chains by adopting these definitions in appendix D, and our partial reward score assesses the correctness for one step beyond the current partial chain context. Let $a^{[l]} = [a^1, \ldots, a^l]$, be partial reasoning chain up to step $l$, and $a^{-[l]} = [a^{l+1}, \ldots, a^L]$ be the completion of the partial reasoning chain from step $l + 1$. We define the intermediate reward by $e^{\eta r(a^{[l]}, x)} = \mathbb{E}_{a^{-[l]} \sim \pi_0} e^{\eta r(a,x)}$. Thus, we assume we have prompt $x$ and the first $a^{[l]}$ steps, we get the derivation for the remaining $a^{-[l]}$:

$$\pi_*(a^{-[l]}|x, a^{[l]}) = \frac{\pi_0(a^{-[l]}|x, a^{[l]}) e^{\eta r(a,x)}}{\mathbb{E}_{a^{-[l]} \sim \pi_0(\cdot|x, a^{[l]})} e^{\eta r(a,x)}},$$

and given prompt $x$ and we obtain the derivation for $a^{[l]}$:

$$\pi_*(a^{[l]}|x) = \frac{\pi_0(a^{[l]}|x) e^{\eta r(a^{[l]}, x)}}{\mathbb{E}_{a^{[l]} \sim \pi_0} e^{\eta r(a^{[l]}, x)}}. \tag{1}$$

$$r(a^{[l]}, x) = \frac{1}{\eta} \ln \mathbb{E}_{a^{-[l]} \sim \pi_0} e^{\eta r(a,x)}. \tag{2}$$

The partial reward according to this formula is our definition of entropy regularized process reward. We can optimize partial reasoning step $\pi(a^{[l]}|x)$ using this process reward.

An important property of this formulation is that our process reward model can be computed by using the reference policy $\pi_0$ to generate the completion. In comparison, in traditional RL, the reward depends on the optimal policy that generates the completion. Therefore in the traditional RL, one has to learn reward and policy simultaneously. This is not necessary in our entropy-based approach. As one can see, equation 2 employs soft optimism over paths generated by the reference policy. More generally, the reward can be computed using updated policy including the optimal policy. In this case, the equivalent formula becomes

$$r(a^{[l]}, x) = -\frac{1}{\eta} \ln \mathbb{E}_{a^{-[l]} \sim \pi_*} e^{-\eta r(a,x)}, \tag{3}$$

where we have used the following fact to express equation 2 using $\pi_*$:

$$\pi_0(a^{-[l]}|x,a^{[l]}) = \frac{\pi_*(a^{-[l]}|x,a^{[l]})e^{-\eta r(a,x)}}{\mathbb{E}_{a^{-[l]}\sim\pi_*}e^{-\eta r(a,x)}}.$$

Intuitively, equation 3 implements soft pessimism over paths generated by the optimal policy. It can be shown that the reward of this model is equivalent to the reward computed using the reference policy in equation 2. As we have already pointed out, the fact that entropy-regularized process reward model can be computed using the reference policy is a key advantage of entropy regularization approach. In this paper, we will use equation 2 to compute our process reward models, and demonstrate its advantage over previous approaches empirically.

We note that the process reward has two formulations: when completions are generated from the initial policy $\pi_0$, it is soft-max, and when completions are generated from the optimal policy, it is soft-min. If one trains LLM and reward model simultaneously using examples generated by the updated LLM, then one needs to gradually turn from soft max to soft min during the training process. For ease of understanding, we provide intuitive explanations of our approach's soft-max and soft-min perspectives in Appendix E, along with a detailed analysis of the theoretical advantages of entropy regularization for process rewards.

## 4 Experimental Settings

We incorporated a figure in Appendix F to further clarify our experiment settings.

**Process Reward Construction**  We conduct experiments using two widely adopted mathematical datasets: GSM8K (Cobbe et al., 2021) and MATH (Hendrycks et al., 2021). Our process reward data is generated and labeled through an automatic labeling approach with entropy regularization, similar to Math-Shepherd (Wang et al., 2024), and is inspired by Monte Carlo Tree Search (MCTS).

In this automatic labeling process, the quality of each step is defined by its potential to deduce the correct final answer. To enhance data diversity, we use a generator to sample 15 solutions per problem, each including step-by-step reasoning. For each step, we employ a completer to generate 16 possible paths leading to the final answer. The quality of a given step is then assessed based on the number of sampled paths that produce the correct final answer. The process reward score for each step is computed using Equation 2, based on the initial policy $\pi_0$. Specifically, the reward function $r(a,x)$ assigns a value of 1 if the step leads to the correct final answer and 0 otherwise.

**Process Reward Sampling**  We conduct experiments using Mistral-7B (Jiang et al., 2023), fine-tuned on the MetaMath dataset (Yu et al., 2024) which we denote as Mistral-MetaMath-7B, and DeepSeek-math-7B-instruct (Shao et al., 2024). These models serve as the generator and completer for data collection, using the method introduced in the paragraph above. For each model, we collect approximately 260K data points and 1.5 million step annotations. Our reward models are trained using Llama-3.1-8B (Dubey et al., 2024).

The reward models are trained on the following datasets:

- **Mistral-data:** Process reward training data collected using Mistral-MetaMath-7B (Yu et al., 2024) as both the generator and completer.
- **DeepSeek-data:** Process reward training data collected using DeepSeek-math-7B-instruct (Shao et al., 2024) as both the generator and completer.

In our main experiments (Section 5), we train the reward models using Llama-3.1-8B on both Mistral-data and DeepSeek-data. Additionally, in Section 6.1, we investigate reward model training at different scales by using Llama-3.2-1B-Instruct and Llama-3.2-3B-Instruct (Dubey et al., 2024) on Mistral-data.

**Process Reward Model Training**  The reward models are trained with a global batch size of 64, a learning rate of 1e-6, a maximum sequence length of 512, and a single epoch using four H100 GPUs. We perform hyperparameter tuning on the learning rate, exploring values from $\{5e^{-6},2e^{-6},1e^{-6},5e^{-7}\}$. During

the automatic labeling process, we set $\eta = 2$ for Mistral data, and $\eta = 10$ for DeepSeek data in our main experiments. We also conduct hyperparameter tuning for $\eta$ across the values $\{0.1, 1, 2, 5, 8, 10, 15\}$.

**Process Reward Model Evaluation**   We evaluate our reward models using the full test set of GSM8K and the MATH500 dataset introduced by Lightman et al. (2023). Our evaluation follows a best-of-N strategy. For each problem in the test set, we randomly sample N solutions from the generators. Each step within a generated response is scored by the reward model. The final answer is selected as the response with the highest overall score. The score for each response is determined by the lowest-scoring step within it.

**Baseline Settings and Training Details**   To evaluate the effectiveness of our method, we compare our models against three baselines: the hard-label process reward model (Hard-label PRM), the vanilla soft-label process reward model (Soft-label PRM), and the outcome reward model (ORM).

- **Hard-label PRM:** A reasoning step is labeled as good if it ultimately leads to the correct final answer.
- **Soft-label PRM:** The step score is approximated by the probability of reaching the correct final answer.
- **ORM:** Assigns a single overall score to the entire reasoning process indicating correct or not.

The reward calculation methods for these models are illustrated in Figure 1. We train the Hard-label PRM, Soft-label PRM, and ORM using the **next token prediction**, also known as auto-regressive training. Specifically, PRM is trained to predict the probability of a special token $'+'$ after each reasoning step, while ORM predicts this token at the end of the response. The reward models are optimized using cross-entropy loss between the predicted and ground-truth rewards, producing scores in the range of $(0, 1)$. The dataset format follows Math-Shepherd (Wang et al., 2024). In addition to the next token prediction, we explore regression-based training in Section 6.4. We also explore reward models with various sizes in Section 6.1, and out-of-distribution (OOD) performance with stronger policy models as generators in Section 6.2. For each experiment settings, we repeat 5 times with different seeds and report the average accuracy along with the standard deviation to ensure the reliability of the results.

**Policy Optimization with PRM-Guided Reinforcement Learning**   We conduct experiments to enhance our policy models, **Mistral-MetaMath-7B** and **DeepSeek-Math-7B-Instruct**, using reinforcement learning from human feedback (RLHF). Specifically, we adopt the rejection sampling fine-tuning strategy (Dong et al., 2023) on the MATH training set and the OpenMathInstruct-2 dataset (Toshniwal et al., 2024b). For each question, we generate 16 responses and rank them using our process reward model (PRM). We then select the highest-scoring responses and fine-tune the policy model on this filtered dataset. The fine-tuned policy models are evaluated on GSM8K and MATH using Top-1 accuracy under a zero-shot chain-of-thought (CoT) setting. Additionally, we compare policy models optimized with our reward model against those guided by baseline reward models. We also report the average accuracy and the standard deviation.

## 5   Experimental Results

### 5.1   Auto-Regressive Training Results

We train the reward model with Llama-3.1-8B as the base model. The results of our model compared with several baseline methods are shown in Figure 2 and Figure 3, which are trained in an auto-regressive way. The details are also listed in Table 5 and Table 6 in the Appendix.

**Mistral-MetaMath-7b as the Generator, Reward Models trained on Mistral-data**   When using Mistral-MetaMath-7B as the policy model, we observe that as N increases, the best-of-N accuracy improves for most models. Notably, ER-PRM consistently outperforms all baseline models across all evaluation settings. In general, process reward models (PRMs) outperform outcome reward models (ORMs), demonstrating the effectiveness of stepwise reward modeling. Our method shows consistent improvements, with particularly significant gains on more challenging datasets. Specifically, on MATH, which contains more complex problems than GSM8K, the performance gap widens substantially. This highlights our approach's superior ability to handle difficult mathematical reasoning tasks.

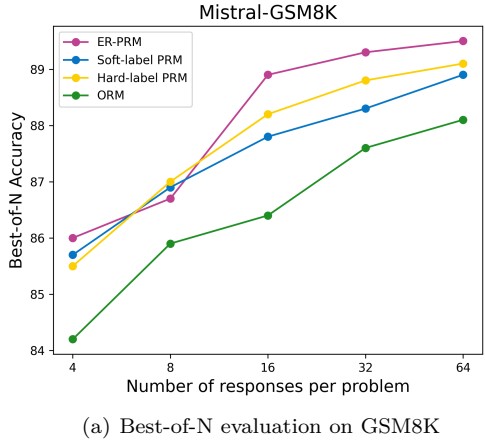

(a) Best-of-N evaluation on GSM8K

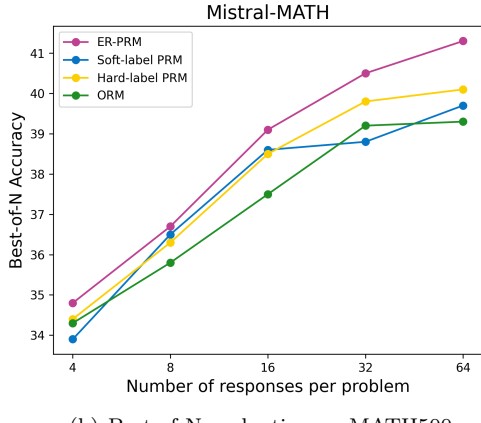

(b) Best-of-N evaluation on MATH500

Figure 2: Best-of-N evaluation results on GSM8K and MATH500 datasets with **Mistral-MetaMath-7b** as the generator. The reward models are trained on **Mistral**-data (GSM8K and MATH).

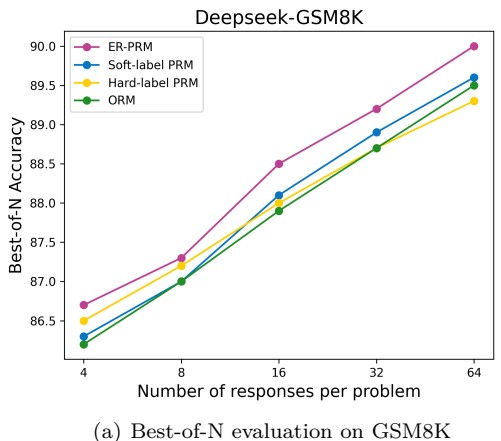

(a) Best-of-N evaluation on GSM8K

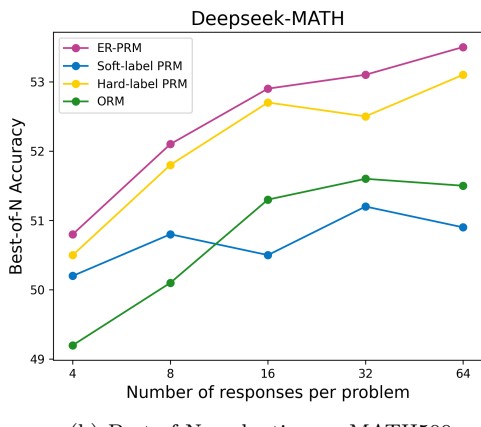

(b) Best-of-N evaluation on MATH500

Figure 3: Best-of-N evaluation results on GSM8K and MATH500 datasets with **DeepSeek-math-7b-instruct** as the generator. The reward models are trained on **DeepSeek**-data.

**DeepSeek-math-7b-instruct as the Generator, Reward Models trained on DeepSeek-data**
When trained on DeepSeek-data and evaluated on DeepSeek-math-7B-Instruct, the results show a significant accuracy boost. ER-PRM consistently outperforms Soft-label PRM and Hard-label PRM on GSM8K and MATH500. However, we observe that on MATH500, the performance improvement from N=4 to N=64 is less pronounced. Additionally, we identify reward hacking issues when using large N. Similar trends are also observed in Section 6.2, where OpenMath2-Llama3.1-8B is used as the policy model. Interestingly, ORM achieves exceptionally strong performance on GSM8K, suggesting that a well-trained ORM with high-quality data can be highly effective.

We also find that Hard-label PRM exhibits strong performance in best-of-N evaluation. In most cases, especially for large N, it surpasses Soft-label PRM. We attribute this to the semi-supervised nature of the Hard-label training process. Since the Markov Decision Process (MDP) labels are only approximations of the process reward rather than precise values, the Hard-label setting helps reduce noise during training.

We note that here the Hard-label setting adopts the same labeling strategy with Math-Shepherd (Wang et al., 2024), trained with the same base model as ER-PRM to ensure a fair comparison. Our model outperforms

Math-Shepherd in nearly all settings on GSM8K and MATH, demonstrating the ability to better rank and select candidate responses. These results highlight the consistency and robustness of our reward models, making them more competent than existing open-source alternatives.

## 5.2 RLHF Results

To verify the effectiveness of our auto-regressive Entropy-Regularized Process Reward Model (ER-PRM), we compare it against the Soft-Label PRM and Hard-Label PRM within the context of RLHF using the RAFT algorithm (Dong et al., 2023). We employ Mistral-MetaMath-7b and DeepSeek-math-7B-instruct as the initial baseline policy models and adopt the reward models trained on Mistral data and DeepSeek data, respectively.

### 5.2.1 Manifestation and Mitigation of Overfitting

To clarify our evaluation methodology, all results reported in this paper were evaluated exclusively on the test sets of GSM8K and MATH, never on training data. During initial experiments, we observed that when using only the original GSM8K and MATH training sets for RLHF, the models showed limited generalization to the test sets, suggesting potential overfitting since these models had already been exposed to these datasets during their original supervised fine-tuning. Our analysis confirmed that Mistral-MetaMath-7B was fine-tuned on Meta-MathQA (which incorporates GSM8K and MATH problems), while DeepSeek-Math-7B-Instruct explicitly included these datasets in its training corpus.

| Models | Methods | GSM8K | MATH |
|--------|---------|-------|------|
| Mistral | Baseline | 77.9 | 28.6 |
| | Soft-Label | 78.5±0.3 | 30.4±0.8 |
| | Hard-Label | 79.0±0.5 | 31.5±0.5 |
| | ER-PRM | **79.7±0.3** | **32.5±0.4** |
| DeepSeek | Baseline | 82.0 | 42.8 |
| | Soft-Label | 82.9±0.4 | 42.8±0.4 |
| | Hard-Label | 82.4±0.3 | 43.8±0.8 |
| | ER-PRM | **83.4±0.3** | **44.7±0.7** |

Table 1: Zero-shot CoT evaluation for policy model **Mistral-MetaMath-7b** and **DeepSeek-math-7b-instruct** improved via Rejection Sampling Fine-tuning. Mistral and DeepSeek in the table denote Mistral-MetaMath-7b and DeepSeek-math-7b-instruct.

To address this overfitting challenge while preserving the consistency of model behaviour, we incorporated 12,500 novel questions from the OpenMathInstruct-2 dataset (Toshniwal et al., 2024a) into our RLHF training data to conduct a blended training. This dataset contains synthetic mathematical problems and rationales that are not present in the original training data of our policy models. For each question, the initial policy model generated 16 candidate rationales (the number of candidates was selected by grid search), which were then evaluated by the ER-PRM and baseline PRMs to select the highest-scoring rationale for RLHF training. The OpenMathInstruct-2 data was used only for training, never for evaluation.

## 5.3 Performance Evaluation

Table 1 summarizes the performance improvements achieved using RLHF enhanced via ER-PRM compared to baseline PRMs. ER-PRM outperforms the baselines, surpassing Soft-Label PRM by 0.8% and 2.7% on GSM8K and MATH, respectively, and Hard-Label PRM by 0.5% and 0.8% for the Mistral model. We observe even larger improvements for the DeepSeek model compared to the baselines, highlighting ER-PRM's superior ability to guide the more potent policy model and its robustness in handling more complex task settings. Additionally, we find that the Hard-Label setting consistently outperforms the Soft-Label setting in RLHF, with the surprisingly strong performance of Hard-Label indicating its feasibility. This is a notable discovery, differing from the findings of Luo et al. (2024).

In general, our RLHF approach achieved superior performance in evaluating the policy model on GSM8K and MATH test sets. This improvement can be attributed to our ER-PRM, which consistently demonstrates 1-2% higher best-of-N accuracy in selecting data from both in-domain and out-of-domain datasets compared to the baselines. This highlights the stability and effectiveness of our entropy-regularized reward formulation in selecting accurate rationales from varying quality data sources.

| Best-of-N | GSM8K | | | MATH500 | | |
|---|---|---|---|---|---|---|
| | Ours | Soft-label PRM | Hard-label PRM | Ours | Soft-label PRM | Hard-label PRM |
| N=4 | 82.1±0.2 | **82.2±0.3** | 81.9±0.2 | **32.3±0.5** | 32.0±0.6 | 31.5±0.5 |
| N=8 | **82.6±0.1** | 82.3±0.2 | 82.2±0.4 | **34.2±0.5** | 33.7±0.5 | 33.5±0.4 |
| N=16 | **83.5±0.3** | 83.2±0.2 | 82.8±0.3 | **37.0±0.6** | 35.4±0.6 | 36.3±0.7 |
| N=32 | **84.1±0.3** | 83.6±0.5 | 83.8±0.4 | **36.4±0.8** | 35.9±0.7 | 36.2±1.1 |
| N=64 | **84.0±0.3** | 83.8±0.4 | 83.9±0.4 | **36.3±0.9** | 35.5±0.9 | 35.9±1.3 |
| Top1@acc | 78.09 | | | 29.4 | | |

Table 2: Best-of-N evaluation for policy model **Mistral-MetaMath-7b**, The reward models are trained from **Llama-3.2-1B-Instruct** on **Mistral** data (GSM8K and MATH).

## 6 Analysis

### 6.1 Reward Models with Various Sizes

In addition to training the reward model with Llama-3.1-8B, we also explore training on various sizes. In this section, we train the reward model with Llama-3.2-3B-Instruct and Llama-3.2-1B-Instruct (Dubey et al., 2024). We hope to increase the performance of smaller-size models so we use the instruct-version for them instead of base models. We train both of them using the Mistral data.

**Experiments with Llama-3.2-1B-Instruct** The experiment results are shown in Table 2. Although we change into models of smaller sizes, the best-of-N accuracy still outperforms the Top 1 accuracy by a large margin. For example, for the 1B size model, the best-of-4 accuracy outperforms Top 1 accuracy by 4% and the best-of-64 accuracy outperforms by 5∼6% on GSM8K dataset. We observe that the performance gain from best-of-4 to best-of-64 for the 1B model is much smaller than the 8B model shown in Table 5, which has 4∼6% performance gain while 1B model has only 2∼3% performance gain. The comparison demonstrates that larger models have higher chances of selecting the gold answer from candidates when $N$ is large. We also discover that the 1B model suffers from severe reward hacking problems on the MATH test set. For ER-PRM and Hard-label PRM, they both achieve an accuracy of 37.0% on best-of-16 evaluation. However, their accuracy drops about 2% on best-of-64 evaluation, showing that they are distracted by the wrong answers and could not effectively select the gold responses. The reward hacking problem becomes less severe for the easy dataset GSM8K, which probably due to the less number of wrong candidate answers.

**Experiments with Llama-3.2-3B-Instruct** The experiment results are shown in Table 3. For the 3B size model, it consistently outperforms the 1B size model, showing that training with larger-size reward models would generally achieve better performance. The 3B reward model on MATH shows a competent performance as the 8B reward model (Table 5), demonstrating the feasibility of training small-size but powerful models. We also discover that the reward hacking problem is largely mitigated for the 3B size model, as the best-of-N accuracy keeps increasing on MATH as $N$ increases, and does not drop much on GSM8K from best-of-32 to best-of-64. However, we still notice that on the GSM8K test set, the performance gain from $N = 4$ to $N = 64$ is still little compared to the 8B case. We may conclude that a larger model size reward model would generally have a larger performance gain when $N$ increases.

In both 1B and 3B size settings, ER-PRM consistently outperforms the Soft-label PRM and Hard-label PRM, demonstrating the effectiveness of our methods across various model sizes.

### 6.2 Out-of-Distribution Performance of Process Reward Models

In this subsection, we examine the OOD performance of the PRMs. The DeepSeek-Math-7B-RL (Shao et al., 2024) and OpenMath2-Llama3.1-8B (Toshniwal et al., 2024b) are used as the policy models for generation, and the reward models are used for best-of-N evaluation.

**DeepSeek-Math-7B-RL as the policy model** We firstly use DeepSeek-Math-7B-RL as the policy model for best-of-N generation. We train the reward model from Llama-3.1-8b on **Mistral data**. The

| Best-of-N | GSM8K | | | MATH500 | | |
|---|---|---|---|---|---|---|
| | **Ours** | **Soft-label PRM** | **Hard-label PRM** | **Ours** | **Soft-label PRM** | **Hard-label PRM** |
| N=4 | **83.6±0.3** | 83.3±0.3 | 83.3±0.3 | **35.6±0.6** | 35.2±0.4 | 35.3±0.7 |
| N=8 | **84.1±0.4** | 83.7±0.5 | 83.5±0.4 | **37.7±0.8** | 37.1±0.8 | 37.4±0.5 |
| N=16 | **84.5±0.3** | 84.0±0.2 | 83.9±0.4 | **38.9±0.6** | 38.2±0.7 | 38.3±0.6 |
| N=32 | **85.0±0.2** | 84.6±0.4 | 84.4±0.5 | **40.8±1.1** | 39.6±0.9 | 40.1±0.8 |
| N=64 | **85.4±0.4** | 85.0±0.4 | 84.8±0.3 | **41.9±1.0** | 40.2±0.8 | 41.1±0.8 |
| Top1@acc | 78.09 | | | 29.4 | | |

Table 3: Best-of-N evaluation for policy model **Mistral-MetaMath-7b**, The reward models are trained from **Llama-3.2-3B-Instruct** on **Mistral** data (GSM8K and MATH).

results are shown in Figure 4 and Table 10. The accuracy increases slowly when N increases from 4 to 64 on GSM8K. And the increment on the MATH dataset is also slower than in the case where we use Mistral as the policy model. We observe the performance drop for ORM reward models, indicating its instability for out-of-distribution generated data. The reason for the smaller performance gain is that, our process reward data is generated from a weaker model Mistral-MetaMath-7b. The reward model might only understand the behavior of the weak model. For certain steps where the weak model fails to produce the correct answer, the reward model is likely to assign these steps a low score. However, a stronger model may be capable of deriving the correct answer in such cases.

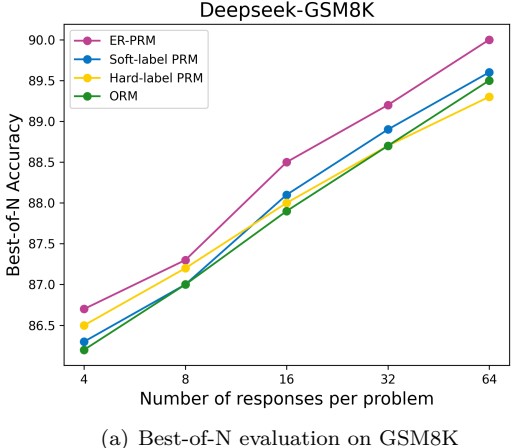

(a) Best-of-N evaluation on GSM8K

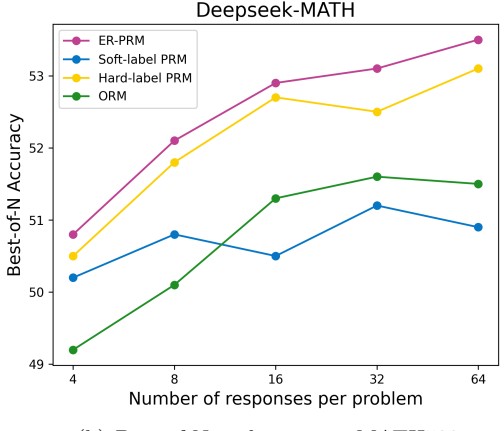

(b) Best-of-N evaluation on MATH500

Figure 4: Best-of-N evaluation results on GSM8K and MATH500 datasets with **DeepSeek-Math-7B-RL** as the generator. The Hard-label PRM is the same setting as Math-Shepherd.

**OpenMath2-Llama3.1-8B as the policy model**  We want to examine the reward model performance paired with one of the most powerful open-source policy models fine-tuned from open-source data. Especially, we hope to lift the performance on MATH, which is difficult for most of the open-source models. We use OpenMath2-Llama3.1-8B (Toshniwal et al., 2024b) as the generator, which is fine-tuned from 14 million mathematical problems and excellent in mathematical reasoning. It achieves a Top 1 accuracy of 66.2% of the MATH500 test set. We evaluate the best-of-N performance on the MATH500 test dataset. For each question, we use OpenMath2-Llama3.1-8B to generate N candidate solutions and use the reward model to select the response. We use the reward model trained from Llama-3.1-8B on **DeepSeek-data**.

The results are shown in Table 4. We first observe a satisfactory performance gain compared to the Top 1 accuracy (66.2%) for all 3 reward models. It could increase the accuracy by more than 4%. The results show that even the reward model trained with data from a weaker model, it could still guide the decoding process during the inference time. We also observe the reward hacking problem when we use the weak reward model to improve the inference of the strong policy model. As shown in the table, there is not much performance

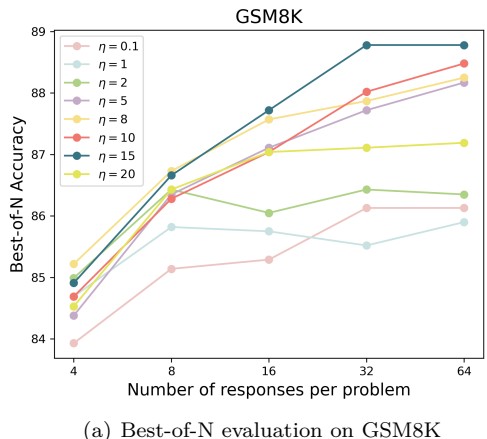

(a) Best-of-N evaluation on GSM8K

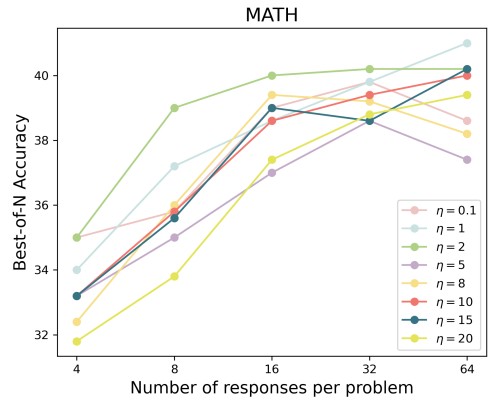

(b) Best-of-N evaluation on MATH500

Figure 5: Best-of-N evaluation results on GSM8K and MATH500 datasets with **Mistral-MetaMath-7b** as the generator with different $\eta$. The reward models are trained on **Mistral** data on GSM8K and MATH **separately**.

gain from $N = 4$ to $N = 64$. And $N = 64$ does not perform the best. We discover that the ER-PRM method still improves the policy model most, demonstrating the feasibility of our method when implemented with strong models.

### 6.3 Impact of Entropy Regularization

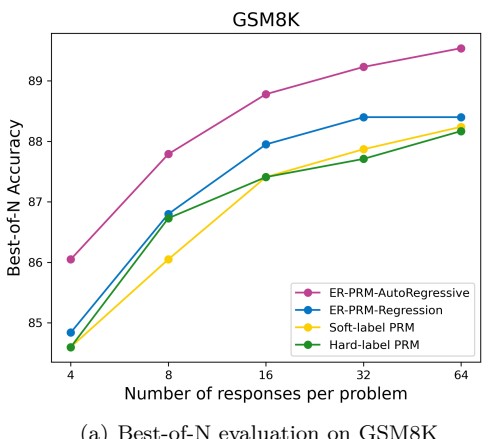

(a) Best-of-N evaluation on GSM8K

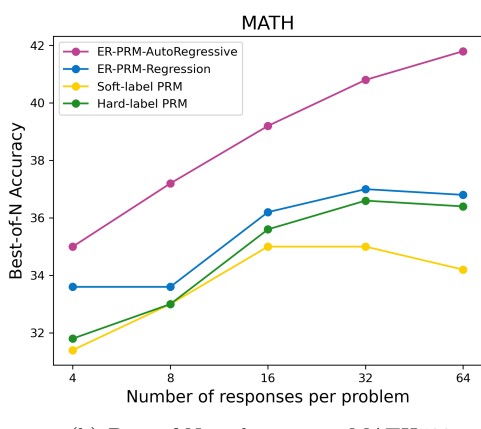

(b) Best-of-N evaluation on MATH500

Figure 6: Best-of-N evaluation results on GSM8K and MATH500 datasets with **Mistral-MetaMath-7b** as the generator. The reward models are trained using regression. They are also compared with the auto-regressive training strategy. The reward models are all trained on **Mistral** data.

To better interpret our entropy-regularized method, we change the hyper-parameter $\eta$ to regularize the penalty of the entropy during automatic process reward labeling. We train the reward model on GSM8K and MATH dataset **separately**, instead of training on **both** datasets in the main experiments. We then evaluate our models using the best-of-N strategy on each dataset. Note that when $\eta \to 0$, the labeling strategy is vanilla soft labeling. When $\eta \to \infty$, the labeling strategy is hard labeling. We show the performance change as $\eta$ changes, and intend to find the optimal $\eta$ for these two datasets.

The result is shown in Figure 5 for best-of-N performance on each $\eta$. We observe that there is a peak performance for each dataset and then the performance drops when $\eta$ is too large or small. The experiments show that $\eta = 1$ or $\eta = 2$ is the best for hard dataset MATH. $\eta = 15$ is the best for easy dataset GSM8K. Larger $\eta$ represents a larger penalty on the entropy for the process rewards where they are pushed higher and closer to the hard label estimation, while smaller $\eta$ represents a small penalty on the entropy for the process rewards. We conclude that to achieve the best performance on automatic process reward labeling, a more fine-grained estimation of the reward scores should be applied. Simply applying hard label estimation may introduce too much noise for reward model training. We also notice that vanilla soft label estimation may not be an optimal strategy, as the performance is relatively bad when $\eta$ is small ($\eta = 0.1$) for both datasets.

Considering the difficulty of the GSM8K and MATH datasets, We may need a larger entropy penalty for the process reward for the hard dataset, and a relatively smaller entropy penalty estimation for the easy dataset.

### 6.4 Regression Training Strategy

In addition to training the reward model in an auto-regressive way, we also explore the reward models with the regression training strategy, where we train the model to output a score directly given a problem and current steps. The best-of-N evaluation results are shown in Figure 6.

The performance of best-of-N increases as N becomes larger. Our reward model still outperforms the Soft-label and Hard-label settings. We also identify it as a method to train reward models. However, we notice that auto-regressive training could yield better results for reward models than regression training. On both GSM8K and MATH datasets, the best-of-N accuracies are consistently lower than the models trained using the auto-regressive strategy. Especially on MATH, there is a significant gap of the accuracy between the two training methods. One hypothesis is that auto-regressive training integrates the steps, reducing the total number of train-

| Best-of-N | MATH | | |
|---|---|---|---|
| | Ours | Soft-label PRM | Hard-label PRM |
| N=4 | 69.1±0.4 | 67.5±0.6 | **69.2±0.3** |
| N=8 | **70.4±0.6** | 69.1±0.6 | 69.2±0.5 |
| N=16 | **70.5±0.5** | 69.3±0.8 | 69.7±0.6 |
| N=32 | **70.3±0.4** | 68.8±0.7 | 69.6±0.7 |
| N=64 | **70.4±0.6** | 68.3±0.5 | 68.8±0.4 |

Table 4: Best-of-N evaluation on the MATH500 test set. The reward models are trained from **Llama-3.1-8B** on **DeepSeek** data. The policy model for generation is **OpenMath2-Llama3.1-8B**.

ing steps, which in turn leads to less forgetting. For the MATH dataset, since there are more reasoning steps per question, splitting them into multiple training samples increases the difficulty for the reward models to learn. It is also possible that the reasoning steps in auto-regressive training are continuous and coherent, allowing the reward model to better learn the process of assigning rewards.

## 7 Conclusion

In this paper, we demonstrate the potential of process reward models (PRMs) in enhancing the performance of large language models (LLMs) on complex mathematical reasoning tasks. By leveraging the fine-grained feedback provided by PRMs, we achieve a notable improvement over traditional outcome reward models (ORMs), as PRMs better capture the multi-step reasoning required for such tasks. To enhance the process reward model, we formulate the multi-step reasoning in the entropy perspective and then derive the process reward score by a new aggregation method. The effectiveness of our automatic labeling process rewards, coupled with entropy-regularized reinforcement learning techniques, highlights the importance of balancing reward optimization with policy stability. Our model outperforms baseline ORMs and PRMs by a large margin on the MATH and GSM8K benchmarks. Furthermore, we apply our process reward model to improve the policy model under RLHF experiments and observe a significant improvement after rejection sampling.

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

## A  Ethical Statements and Limitations

Our work enhances mathematical reasoning in LLMs using an entropy-regularized process reward model (ER-PRM), and we have carefully considered ethical concerns in data usage and algorithmic design. For data sources, our experiments rely on publicly available human preference datasets, ensuring compliance with their respective licenses. These datasets do not involve privacy issues, and we adhere to ethical guidelines in their use. Algorithmically, while ER-PRM improves reasoning stability, reward modeling can influence LLM behavior, potentially reinforcing unintended biases in problem-solving approaches. Additionally, if not carefully managed, preference learning could lead to overfitting on certain reasoning styles, limiting model generalization. To mitigate these risks, we encourage rigorous testing and diverse dataset construction. Lastly, advanced reasoning models pose potential misuse risks, such as academic dishonesty or exploitation in high-stakes decision-making. We recommend ongoing safety evaluations and responsible deployment to ensure robustness and fairness in real-world applications.

ER-PRM has several limitations. High computational cost may hinder scalability, and overfitting risks require careful tuning, particularly in DeepSeek-Math-7B-Instruct. Our approach is mathematics-focused, and its generalization to other domains remains untested. Finally, reward aggregation mechanisms could benefit from more interpretability. Addressing these issues will improve the model's robustness and applicability.

## B  Technical Lemma

**Lemma 1 (Solution of KL-regularized Optimization)** *Given a loss functional with respect to $p(\cdot|x)$, written as*

$$\mathbb{E}_{w \sim p(\cdot)} \Big[ -U(w) + \eta D_{\mathrm{KL}}\big(p(\cdot), p_0(\cdot)\big) \Big]$$
$$= \eta D_{\mathrm{KL}}\Big( p(\cdot), p_0(\cdot) \exp\Big(\frac{1}{\eta} U(\cdot)\Big) \Big)$$
$$- \eta \cdot \log \underbrace{\mathbb{E}_{w \sim p_0(\cdot)} \exp\Big(\frac{1}{\eta} U(w)\Big)}_{C_r},$$

*where the minimizer of the loss functional is $p^*(w) = \frac{1}{C_r} p_0(w) \exp\Big(\frac{1}{\eta} U(w)\Big)$, also known as Gibbs distribution.*

See Proposition 7.16 and Theorem 15.3 of Zhang (2023) for a detailed proof.

## C  Best-of-N Evaluation Results in Tables

In this section, we list the performance of the best-of-N evaluation of ER-PRM and other baselines in Tables. The results for the reward models with policy model **Mistral-MetaMath-7b** are shown in Table 5, which corresponds to Figure 2. The results for the reward models with policy model **Deepseek-Math-7B-Instruct** are shown in Table 6, which corresponds to Figure 3.

The results for Section 6.3 in Analysis are shown in Table 7 and Table 8, corresponding to the GSM8K and MATH datasets. The results for Section 6.4 are shown in Table 9.

## D  Definition of Step, Partial Chain, and Chain

**Step:**
A step refers to the smallest unit of operation in a reasoning process. Each step performs a single operation (e.g., a single calculation or logical deduction) and represents one incremental action within the overall reasoning process.

| Best-of-N | GSM8K | | | | MATH500 | | | |
|---|---|---|---|---|---|---|---|---|
| | Ours | Soft-label PRM | Hard-label PRM | ORM | Ours | Soft-label PRM | Hard-label PRM | ORM |
| N=4 | **86.0±0.3** | 85.7±0.2 | 85.5±0.4 | 84.2±0.6 | **34.8±0.8** | 33.9±0.5 | 34.4±0.7 | 34.3±0.5 |
| N=8 | 86.7±0.4 | 86.9±0.3 | **87.0±0.3** | 85.9±0.7 | **36.7±0.7** | 36.5±0.8 | 36.3±0.6 | 35.8±0.6 |
| N=16 | **88.9±0.4** | 87.8±0.4 | 88.2±0.3 | 86.4±0.5 | **39.1±1.1** | 38.6±0.6 | 38.5±0.9 | 37.5±0.6 |
| N=32 | **89.3±0.3** | 88.3±0.3 | 88.8±0.3 | 87.6±0.7 | **40.5±0.8** | 38.8±0.7 | 39.8±0.6 | 39.2±0.5 |
| N=64 | **89.5±0.3** | 88.9±0.3 | 89.1±0.2 | 88.1±0.6 | **41.3±0.6** | 39.7±0.8 | 40.1±0.8 | 39.3±0.6 |
| Top1@acc | 78.09 | | | | 29.4 | | | |

Table 5: Best-of-N evaluation for policy model **Mistral-MetaMath-7b**. The reward models are trained using **auto-regressive** strategy from **Llama-3.1-8B** on **Mistral** data (GSM8K and MATH).

| Best-of-N | GSM8K | | | | MATH500 | | | |
|---|---|---|---|---|---|---|---|---|
| | Ours | Soft-label PRM | Hard-label PRM | ORM | Ours | Soft-label PRM | Hard-label PRM | ORM |
| N=4 | **86.7±0.3** | 86.3±0.4 | 86.5±0.4 | 86.2±0.3 | **50.8±0.4** | 50.2±0.9 | 50.5±0.6 | 49.2±0.5 |
| N=8 | **87.3±0.5** | 87.0±0.4 | 87.2±0.3 | 87.0±0.4 | **52.1±0.5** | 50.8±0.7 | 51.8±0.7 | 50.1±0.5 |
| N=16 | **88.5±0.5** | 88.1±0.3 | 88.0±0.3 | 87.9±0.3 | **52.9±0.3** | 50.5±1.0 | 52.7±0.3 | 51.3±0.4 |
| N=32 | **89.2±0.3** | 88.9±0.4 | 88.7±0.6 | 88.7±0.4 | **53.1±0.5** | 51.2±0.9 | 52.5±0.5 | 51.6±0.5 |
| N=64 | **90.0±0.4** | 89.6±0.5 | 89.3±0.4 | 89.5±0.6 | **53.5±0.5** | 50.9±0.8 | 53.1±0.4 | 51.5±0.6 |
| Top1@acc | 82.03 | | | | 42.8 | | | |

Table 6: Best-of-N evaluation for policy model **DeepSeek-Math-7B-Instruct**, The reward models are trained from **Llama-3.1-8B** on **DeepSeek** data (GSM8K and MATH).

| | GSM8K | | | | | | | |
|---|---|---|---|---|---|---|---|---|
| **Best-of-N** | $\eta = 0.1$ | $\eta = 1$ | $\eta = 2$ | $\eta = 5$ | $\eta = 8$ | $\eta = 10$ | $\eta = 15$ | $\eta = 20$ |
| N=4 | 83.93 | 84.69 | 84.99 | 84.38 | 85.22 | 84.69 | 84.91 | 84.53 |
| N=8 | 85.14 | 85.82 | 86.43 | 86.35 | 86.73 | 86.28 | 86.66 | 86.43 |
| N=16 | 85.29 | 85.75 | 86.05 | 87.11 | 87.57 | 87.04 | 87.72 | 87.04 |
| N=32 | 86.13 | 85.52 | 86.43 | 87.72 | 87.87 | 88.02 | 88.78 | 87.11 |
| N=64 | 86.13 | 85.90 | 86.35 | 88.17 | 88.25 | 88.48 | 88.78 | 87.19 |

Table 7: Best-of-N evaluation for different $\eta$ values on GSM8K dataset. The reward models are trained from **Llama-3.1-8B** solely on GSM8K.

| | MATH | | | | | | | |
|---|---|---|---|---|---|---|---|---|
| **Best-of-N** | $\eta = 0.1$ | $\eta = 1$ | $\eta = 2$ | $\eta = 5$ | $\eta = 8$ | $\eta = 10$ | $\eta = 15$ | $\eta = 20$ |
| N=4 | 35.0 | 34.0 | 35.0 | 33.2 | 32.4 | 33.2 | 33.2 | 31.8 |
| N=8 | 35.8 | 37.2 | 39.0 | 35.0 | 36.0 | 35.8 | 35.6 | 33.8 |
| N=16 | 39.0 | 38.6 | 40.0 | 37.0 | 39.4 | 38.6 | 39.0 | 37.4 |
| N=32 | 39.8 | 39.8 | 40.2 | 38.6 | 39.2 | 39.4 | 38.6 | 38.8 |
| N=64 | 38.6 | 41.0 | 40.2 | 37.4 | 38.2 | 40.0 | 40.2 | 39.4 |

Table 8: Best-of-N evaluation for different $\eta$ values on MATH dataset. The reward models are trained from **Llama-3.1-8B** solely on MATH.

| Best-of-N | GSM8K | | | MATH500 | | |
|---|---|---|---|---|---|---|
| | Ours | Soft-label PRM | Hard-label PRM | Ours | Soft-label PRM | Hard-label PRM |
| N=4 | 84.84 | 84.60 | 84.60 | 33.6 | 31.4 | 31.8 |
| N=8 | 86.80 | 86.05 | 86.73 | 33.6 | 33.0 | 33.0 |
| N=16 | 87.95 | 87.41 | 87.41 | 36.2 | 35.0 | 35.6 |
| N=32 | 88.40 | 87.87 | 87.71 | 37.0 | 35.0 | 36.6 |
| N=64 | 88.40 | 88.24 | 88.17 | 36.8 | 34.2 | 36.4 |

Table 9: Best-of-N evaluation for policy model **Mistral-MetaMath-7b**, The reward models are trained using **regression** strategy from **Llama-3.1-8B** on **Mistral** data (GSM8K and MATH).

| Best-of-N | GSM8K | | | | MATH500 | | | |
|---|---|---|---|---|---|---|---|---|
| | Ours | Soft-label PRM | Hard-label PRM | ORM | Ours | Soft-label PRM | Hard-label PRM | ORM |
| N=4 | 89.46 | 89.39 | 88.70 | 88.77 | 52.6 | 51.4 | 51.8 | 52.6 |
| N=8 | 89.91 | 89.61 | 88.93 | 89.09 | 55.8 | 54.8 | 55.4 | 55.0 |
| N=16 | 90.22 | 90.22 | 89.76 | 89.01 | 55.6 | 54.8 | 55.4 | 55.2 |
| N=32 | 90.45 | 90.45 | 89.84 | 89.01 | 56.0 | 55.0 | 55.0 | 55.0 |
| N=64 | 90.67 | 90.45 | 89.84 | 88.10 | 57.0 | 55.4 | 55.6 | 53.4 |
| Top1@acc | 88.17 | | | | 51.0 | | | |

Table 10: Best-of-N evaluation for policy model **DeepSeek-Math-7B-RL**. The reward models are trained using **auto-regressive** strategy from **Llama-3.1-8B** on **Mistral** data (GSM8K and MATH).

**Chain:**
A chain represents the complete series of operations that culminate in the final result or target solution. It consists of multiple sequential steps, forming the full reasoning process.

**Partial Chain:**
A partial chain is a subset of a complete chain, consisting of a sequence of intermediate steps leading toward (but not necessarily reaching) the final result. Partial chains are used in our work to compute intermediate reward scores that guide the reasoning process step-by-step.

In our paper, when we refer to "step-level derivation," we are specifically describing the last step reward score associated with partial chains. These scores are step-level reward scores and not chain-level reward scores, as they evaluate the contributions of individual steps (or intermediate sequences) rather than the final output of the entire chain. The distinction is important because step-level scores are critical for guiding the model at each reasoning step, while chain-level scores evaluate the overall success of the complete reasoning process.

# E    Intuitive Understanding and Theoretical Advantages

## E.1    Intuitive Perspective on Entropy-Regularized Process Rewards

We provide an intuitive explanation of entropy-regularized process rewards:

**The Soft-Max View:** When using initial policy model $\pi_0$ (a weaker SFT model like 7B parameter LLM) to generate reasoning paths, our formulation essentially answers the question: "If we continue from this step using our current knowledge, what's the likelihood of reaching a correct answer?" The entropy regularization term (controlled by $\eta$) determines how optimistic this assessment should be. Higher $\eta$ values make the model focus more on the highest-reward paths, creating a "soft-optimistic" evaluation that weights multiple possible futures according to their potential.

**The Soft-Min View:** Conversely, when using an optimal policy model $\pi_*$ (a strong model like GPT-4o), our formula addresses the question: "What's the smallest penalty we would need to apply to make this step as good as other alternatives?" This creates a "soft-pessimistic" evaluation that focuses on the worst-case scenarios but with some forgiveness. This approach employs a more cautious strategy to align with the preferences of a stronger model.

**Mechanics of Reward Aggregation:** To illustrate how our reward aggregation works in practice, consider the step $a^1$ in Figure 1. From this step, we see three possible continuation paths reaching $a^{L1,1}$, $a^{L2,2}$, $a^{L3,3}$. We calculate the correctness of each path individually, and get $y_1 = 1$, $y_2 = 1$, and $y_3 = 0$ since the first two paths derive the correct final answer. Once we get the value of different $y_i$, we are able to calculate the expected reward score using the equation provided in Figure 1.

### E.2 Intuitive Understanding and Theoretical Advantages

Building on this intuitive foundation, we now articulate the key theoretical advantages of our entropy-regularized approach:

(1) **Dual Formulation Flexibility:**

Our process reward approach offers a significant theoretical advantage through its dual formulation. When sampling completions from the initial policy $\pi_0$, our process reward employs a soft-max aggregation as equation equation 2. Conversely, when sampling from the optimal policy $\pi_*$, it implements a soft-min aggregation as equation equation 3.

This duality allows our method to adapt smoothly depending on the sampling policy. When sampling from a weaker initial policy $\pi_0$, the soft-max formulation acts as an optimistic estimator that amplifies high-reward paths, creating stronger learning signals. When sampling from a stronger optimal policy $\pi_*$, the soft-min formulation provides a conservative estimate that focuses on the most reliable paths while maintaining diversity.

(2) **Reward Signal Amplification:**

Traditional methods calculate rewards as simple expectations under the initial policy $\pi_0$: $r(a^{-L}) = \mathbb{E}_{\pi_0}[r(a|a^L)]$. While straightforward, this approach can result in weak signals when $\pi_0$ has a low probability of sampling high-reward trajectories, which is common in complex mathematical reasoning tasks where correct solutions form a small subset of solution space.

Our entropy-regularized formulation addresses this through the exponential transformation $e^{\eta r(a,x)}$, which amplifies the reward signal before expectation. This transformation effectively redistributes probability mass toward high-reward trajectories, making the learning signal stronger and more informative. The subsequent logarithmic transformation maintains the relative ordering of rewards while controlling the scale.

(3) **Independence of Optimal Policy with Single Hyperparameter:**

A key theoretical advantage of our method is that it requires tuning only a single hyperparameter $\eta$ that controls the entropy penalty, providing an elegant way to balance the exploitation between high-reward trajectories and the breadth of policy space.

This contrasts with previous approaches that often require complex weighting schemes or multiple parameters to achieve similar effects. Furthermore, our entropy-regularized process reward can be computed using only the initial policy $\pi_0$, as demonstrated in equation equation 2. This removes a dependency present in previous RL methods, where the optimal policy is needed.

## F   Figure of Experimental Setting

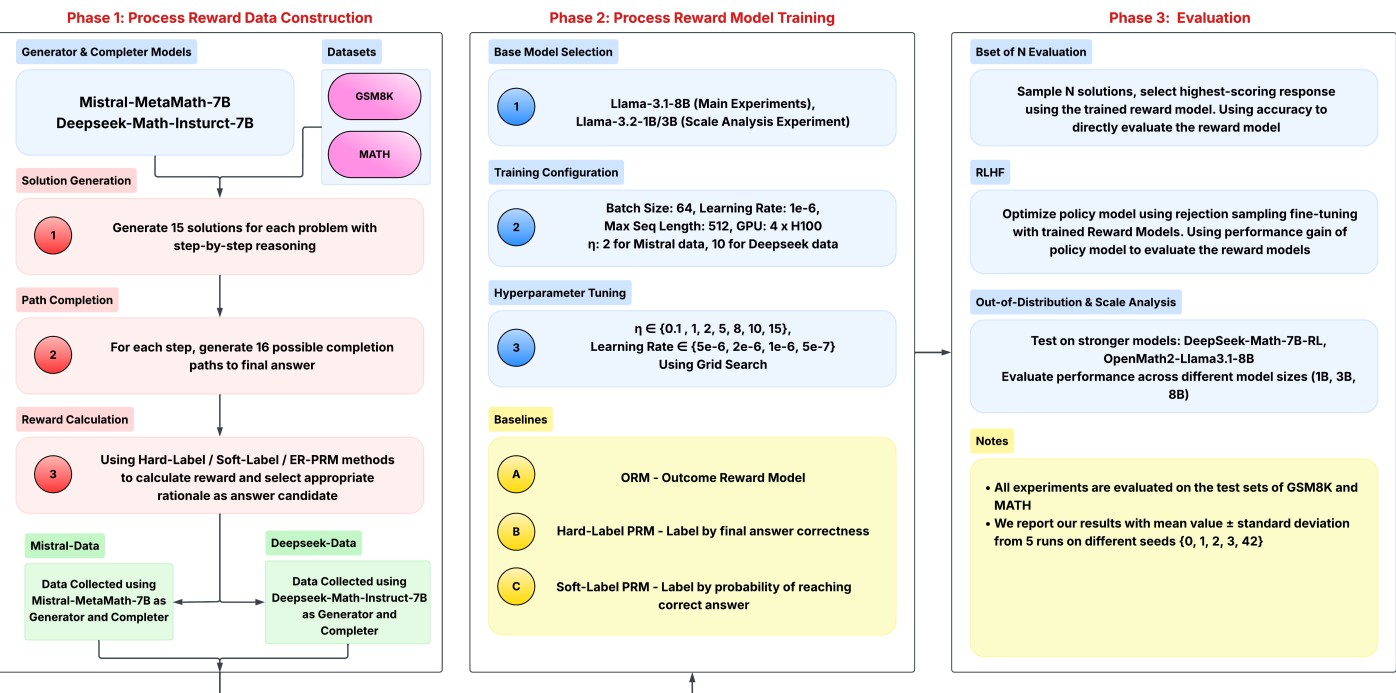

Figure 7: Figurative Illustration of Experimental Setting

