# OpenReview forum: "Entropy-Regularized Process Reward Model"
_TMLR — Accepted by TMLR_

### Review · Reviewer_KBAn · 2025-02-18

**Summary Of Contributions:**

The paper proposes a method for enhancing the mathematical reasoning capabilities of LLMs through an entropy-regularized process reward model (ER-PRM). The authors integrate KL-regularized MDP to balance policy optimization while preventing the policy from deviating too far from its initial distribution. The key contributions include:

1. Entropy-Regularized Process Reward Model (ER-PRM): A new method for labeling process reward scores with entropy regularization, derived from theoretical results.

2. Empirical Validation: Experiments on the MATH and GSM8K benchmarks demonstrate that ER-PRM outperforms existing process reward models, achieving significant improvements in reasoning accuracy.

3. Theoretical Analysis: The paper provides a detailed derivation of the reward construction method and shows that the optimal reward model can be derived from initial policy sampling.

**Audience:**

Yes

**Broader Impact Concerns:**

No concerns. The authors mention the potential for reward hacking and the challenges associated with training reward models, but an explicit Broader Impact Statement would be helpful to highlight these issues.

**Claims And Evidence:**

No

**Requested Changes:**

**Critical**:

- The authors should add a section discussing the ethical implications of their work, including potential biases, fairness, and the broader impact of using entropy-regularized process reward models in LLMs.

- Explain the advantages of the method from a theoretical perspective.

**Recommended**:

- A more detailed analysis of overfitting, including strategies to mitigate it, would strengthen the paper. This could involve discussing regularization techniques or providing additional experiments to demonstrate the model's robustness.

- Expand the Limitations section to discuss computational overhead, scalability, and generalization to non-mathematical tasks.

**Strengths And Weaknesses:**

**Strength**:

1. The integration of entropy regularization into process reward modeling is a fresh approach, addressing the stability of policy optimization in multi-step reasoning tasks.  Derives reward formulations from KL-regularized RL principles, grounding the method in established theory.

2. The paper presents empirical evidence of the effectiveness of ER-PRM. The results on the MATH and GSM8K benchmarks show consistent improvements over existing methods, highlighting the efficacy of the proposed approach. Comprehensive experiments across datasets, model sizes, and training strategies (e.g., auto-regressive vs. regression) demonstrate robustness.

3. The authors provide a solid theoretical foundation for their method, deriving the reward construction method from KL-regularized MDP. This theoretical analysis strengthens the credibility of the proposed model.

4. The paper includes a wide range of experiments, including best-of-N evaluations, RLHF results, and out-of-distribution performance tests. These experiments provide a comprehensive evaluation of the model's capabilities and robustness.

**Weakness**:

1. Section 3 assumes familiarity with KL-regularized RL, which may hinder accessibility. Intuitive explanations or visualizations of the reward aggregation process (soft max/min) are sparse.

2. The paper mentions signs of overfitting in some of the experiments, particularly with the deepseek-math-7b-instruct model. A more detailed analysis of overfitting and strategies to mitigate it would be beneficial.

3. The experimental results are not significant. Could the advantages of ER-PRM be further explained at a theoretical level?

---

> ### Author Response · Authors · 2025-03-04
> **Response to Reviewer KBAn Regarding the Ethical Issues and Limitations**
>
> Dear Reviewer KBAn,
>
> Thank you for your thoughtful review of our paper! We first add the Ethical Implications and the Limitations below
>
> **Ethical Implications**
>
> Our work enhances mathematical reasoning in LLMs using an entropy-regularized process reward model (ER-PRM), and we have carefully considered ethical concerns in data usage and algorithmic design.
>
> For data sources, our experiments rely on publicly available human preference datasets, ensuring compliance with their respective licenses. These datasets do not involve privacy issues, and we adhere to ethical guidelines in their use.
>
> Algorithmically, while ER-PRM improves reasoning stability, reward modeling can influence LLM behavior, potentially reinforcing unintended biases in problem-solving approaches. Additionally, if not carefully managed, preference learning could lead to overfitting on certain reasoning styles, limiting model generalization. To mitigate these risks, we encourage rigorous testing and diverse dataset construction.
>
> Lastly, advanced reasoning models pose potential misuse risks, such as academic dishonesty or exploitation in high-stakes decision-making. We recommend ongoing safety evaluations and responsible deployment to ensure robustness and fairness in real-world applications.
>
> **Limitations**
>
> ER-PRM has several limitations. High computational cost may hinder scalability, and overfitting risks require careful tuning, particularly in deepseek-math-7b-instruct. Our approach is mathematics-focused, and its generalization to other domains remains untested. Finally, reward aggregation mechanisms could benefit from more interpretability. Addressing these issues will improve the model’s robustness and applicability.

---

> ### Author Response · Authors · 2025-03-04
> **Response to Reviewer KBAn Regarding the Weakness**
>
> Dear Reviewer KBAn,
>
> Thank you for your thoughtful review of our paper! We appreciate your recognition of our contributions and the strengths of our approach. Below, we address your concerns and suggestions in detail.
>
> **1. Accessibility of Section 3 and KL-regularized RL**
>
> Thank you for pointing that out and we appreciate your suggestions that could improve the readability of our work. Here is our intuitive way of explaining our theoretical framework:
>
> In section 3, our key insight is that process rewards can be derived in a principled way from this KL-regularized framework. Here's how it works intuitively:
>
> **1.1 The Soft-Max View (when using the initial policy):**
> When we use the initial policy model $\pi_0$ (a weaker SFT model, i.e. various 7B models) to generate reasoning paths, our formula essentially asks: "If we continue from this step using our current knowledge, what's the likelihood of reaching a correct answer?" This creates a "soft-optimistic" evaluation that weights multiple possible futures, which we later refer to as the soft-max approach. This approach guided the model to learn in a bolder and more aggressive way while utilizing the hyperparameter $\eta$ to hedge the risk of drifting away from original policy.
>
> **1.2 The Soft-Min View (when using the optimal policy):**
> When using an optimal policy model $\pi_*$ (a strong SOTA model, i.e. GPT-4o or o1), our formula flips to ask: "What's the smallest penalty we would need to apply to make this step as good as other alternatives?" This creates a "soft-pessimistic" evaluation that focuses on the worst-case scenarios but with some forgiveness, which we later refer to as the soft-min approach. This formulation employs a more cautious, conservative strategy to align the preferences of a strong model which experienced multiple rounds of SFT and RLHF.
>
>
>
> **2. Analysis of Overfitting**
>
> We appreciate you highlighting the need for a more detailed analysis of overfitting observations in our experiments. We agree this deserves further explanation and will expand this analysis in our revision.
>
> Our investigation revealed a specific pattern of overfitting that occurs when applying RLHF to models already trained on the target datasets. Specifically:
>
> **2.1 Data Overlap Analysis:** We carefully analyzed the pre-training data used for both baseline models. We discovered that Mistral-MetaMath-7B was fine-tuned on MetaMathQA, which already incorporates augmented versions of GSM8K and MATH problems. Similarly, deepseek-Math-7B-Instruct explicitly included GSM8K and MATH in its training corpus, as documented in their paper.
>
> **2.2 Manifestation of Overfitting:** When we initially conducted RLHF using the RAFT algorithm with data solely from GSM8K and MATH, all reward models (including baselines and our ER-PRM) produced only marginal improvements on test sets despite showing gains on training data. This pattern is consistent with overfitting - the models were essentially "re-learning" data distributions they had already seen during supervised fine-tuning.
>
> To address this issue while preserving the benefits of our approach, we implemented a data diversity strategy:
>
> **2.3 Introduction of Novel Data:** We incorporated 12,500 additional questions from OpenMathInstruct-2, which contains novel synthetic mathematical problems and rationales not present in the original training data of our policy models.
>
> **2.4 Blended Training Approach:** Rather than completely replacing training data, we used a blended approach that ensures consistency in model behavior while introducing sufficient novelty to prevent overfitting. The improvement in test set performance after implementing this blended approach (as shown in Table 1) confirms our hypothesis about the overfitting mechanism.
>
> In our revision, we will add a discussion of future research directions for addressing overfitting:
>
> We would further investigate the optimal ratio between familiar and novel data in RLHF training and explore the effect of dynamically adjusting $\eta$ values to alleviate the overfitting.
>
> We believe this expanded analysis significantly addresses the reviewer's concern while highlighting how our entropy-regularized approach provides inherent advantages in mitigating overfitting compared to baseline methods.

---

> > ### Author Response · Authors · 2025-03-04
> >
> > **3. Theoretical Advantages of ER-PRM**
> >
> > We deeply thank you for the thoughtful assessment of our work. We particularly appreciate the identification of our theoretical foundation as a strength. We would like to address the concern about the significance of experimental results by elaborating on the theoretical advantages of our approach, which provide a principled explanation for the observed performance improvements.
> >
> > The foundation of our work lies in the principled derivation of process rewards through entropy regularization within the KL-regularized MDP framework. This theoretical grounding provides several distinct advantages that explain our method's improved performance:
> >
> > **3.1 Dual Formulation: Soft-Max and Soft-Min**
> >
> > A key theoretical strength of our approach is the flexibility of a dual formulation. When sampling completions from the initial policy $\pi_0$, our process reward implements soft-max:
> >
> > $r(a^{L}, x) = \frac{1}{\eta}\ln\mathbb{E}_{a^{-[l]} \sim \pi_0}e^{\eta r(a,x)}$.
> >
> > Conversely, when sampling from the optimal policy $\pi_*$, it implements soft-min:
> >
> > $r(a^{L}, x)= -\frac{1}{\eta}\ln\mathbb{E}_{a^{-[l]} \sim \pi^*} e^{-\eta r(a,x)}$.
> >
> > This duality provides theoretical flexibility not present in previous methods. It allows our approach to adapt optimally depending on the policy being used for sampling, facilitating more robust learning across a spectrum of model capabilities.
> >
> > **3.2 Signal Amplification for Smoothing Sparse Rewards**
> >
> > The previous methods normally require sampling from optimal policy model $\pi_*$, but our method can rely on initial weak policy $\pi_0$ by amplifying the sparse rewards with our ER-PRM. In traditional process reward methods, the reward $r(a^{−L})$ is computed as an expectation under the initial policy $\pi_0$ , such that: $r(a^{-L}) = \mathbb{E}_{\pi_0}[r(a|a^{-L})]$.
> >
> > While this approach is straightforward, the sole reliance on the initial policy $\pi_0$ can result in sharp or overly localized reward distributions. This limitation is particularly evident when $\pi_0$ samples from a small, concentrated region, leading to weak or sparse reward signals.
> >
> > Drawing from similar experience with the importance sampling in reinforcement learning, the reward can be estimated as a ratio between the target policy $\pi_*$ and the initial policy $\pi_0$, specifically: $r = \log{\frac{\pi_*}{\pi_0}}$.
> >
> > In our work, we adopt this definition and extend it with exponential transformation $e^{\eta r(a,x)}$ before expectation and logarithmic transformation after. This approach effectively smooths out the reward signal, addressing the shortcomings of traditional process reward models. For instance, when $\pi_0$ reasons over complex mathematical tasks where correct solutions form a small subset of the possible answer space, this amplification is theoretically crucial for effective learning. Our formulation amplifies these weak signals through the ratio $\frac{\pi*}{\pi_0}$, redistributing the reward more effectively across the policy space. This smoothing effect allows for a more robust reward calculation, leading to stronger step-level signals and better optimization during training.
> >
> > **3.3 Simplicity in Hyperparameter and Independence of Optimal Policy**
> >
> > Our method only requires turning a single hyperparameter $\eta$ that controls the entropy penalty. This provides a theoretically elegant way to balance between:
> > 1. Exploitation of high-reward trajectories (higher $\eta$ values)
> > 2. Exploration across the policy space (lower $\eta$ values)
> >
> > Another significant theoretical advantage of our method is that, unlike traditional approaches that depend on sampling from the optimal policy $\pi_*$​, our entropy-regularized process reward can be computed using only the initial policy $\pi_0$​. This removes a dependency problem presented.
> >
> > We believe these theoretical foundations explain why ER-PRM delivers performance gains over baseline methods and demonstrates greater robustness across different experimental settings.

---

### Review · Reviewer_kNgM · 2025-02-21

**Summary Of Contributions:**

This paper introduces entropy-regularized process reward models (ER-PRMs), which are trained similarly to PRMs but have an additional KL-regularized term in their loss function to balance optimizing the reward and staying close to the original policy.

The ER-PRM are evaluated in two scenarios: as inference-time judges to pick the best-of-N response from existing policy models (Mistral-MEtaMath-7B and deepseek-math-7b-instruct); and as training-time reward models to improve policy models in RLHF cycles. The final policy models are then evaluated on GSM8k and MATH and compared to models guided by different baseline reward models.

Overall the proposed method achieves slightly better performance compared to other baselines.

**Audience:**

Yes

**Broader Impact Concerns:**

no concerns

**Claims And Evidence:**

Yes

**Requested Changes:**

- Add standard deviations to results in all tables, and confidence intervals to results in all figures for Claims And Evidence to be evaluated as "yes"

- Be more nuanced in some claims with words like “significantly” being justified only by a few percentage points.

**Strengths And Weaknesses:**

# Strengths

This paper proposes a novel method to train PRMs based on entropy regularization.
The method is tested on various LLMs ranging in size, tested on various datasets ranging in difficulty, and compared to other traditional baselines. Overall the proposed method seems to outperform all other methods, yielding strong PRMs for both inference time filtering and training time filtering during rejection sampling rlhf cycles.
The method is also tested on “unseen” models where the policy model is different than the model that generated training data for the PRM.
Eventually, the authors analyze the effectiveness of their regularization in an ablation study and discuss the auto-regressive -vs- regression training strategies.

# Weaknesses

1. The proposed ER-PRMs outperform other baselines by 1 or 2% points, however without any confidence intervals or standard deviations across multiple runs with different random seeds, this outperformance can be due to random noise. Results must include confidence intervals to see if the difference is statistically significant.

2. The claim (made in Section 5.1) that the proposed method is especially effective on complex tasks is not well supported by evidence. The authors refer to figures 2b and 3b to support this claim. While Figure 2b seems to go in that direction with ER-PRMs outperforming all other baselines (gains from ~39% to ~42% when N=64), the numbers in Figure 3b are much more nuanced. ER-PRM only marginally performs better than Hard-label PRM with gains from ~52.5% to ~53% when N=64.

3. Minor: While it is interesting to see best-of-n approaches performing better than top1 accuracy, the experimental section should not focus too much on this as it is generally expected that any best-of-n approach will yield better performance than top1 accuracy. Since this paper introduces a PRM meant to judge the top N responses, the readers should focus on this. When we focus on other best-of N baseline models, the proposed method is only marginally better (from 0.5 to 2% overall).

# Questions

In section 5.2, the authors mention that they observed that the baseline policy model was overfitting to the training data, resulting in only marginal performance gains after RLHF on the original training set. My question is why did they evaluate on the training set and not the test set?
The authors then highlight that they added 12,500 questions from OpenMAthInstruct-2, but this dataset is not listed in the result table 1. Did they added these examples to the train or test set?

Overall, this makes me question the rest of the results in the paper, was everything evaluated only on the training set and not on hold-out test sets? Further clarifications is needed.

---

> ### Author Response · Authors · 2025-03-05
> **More Experiments and Standard Deviations**
>
> Dear Reviewer kNgM,
>
> We appreciate your thoughtful assessment of our performance results. And we have conducted more experiments with standard deviations included. Specifically, we conducted each experiment 5 times, with random seeds {0,1,2,3,42}.
>
> **Mistral data on Llama-3.1-8b (corresponding to Figure 2 and Table 5)**
>
> | **GSM8K** |             |         |         |
> |-----------------|------------|---------|---------|
> |                | Ours       | Hard    | Soft    |
> | Best-of-4      | **86.0±0.3** | 85.5±0.4 | 85.7±0.2 |
> | Best-of-8      | 86.7±0.4 | 87.0±0.3 | **86.9±0.3** |
> | Best-of-16     | **88.9±0.4** | 88.2±0.3 | 87.8±0.4 |
> | Best-of-32     | **89.3±0.3** | 88.8±0.3 | 88.3±0.3 |
> | Best-of-64     | **89.5±0.3** | 89.1±0.2 | 88.9±0.3 |
>
> | **MATH500** |             |         |         |
> |-----------------|------------|---------|---------|
> |                | Ours       | Hard    | Soft    |
> | Best-of-4      | **34.8±0.8** | 34.4±0.7 | 33.9±0.5 |
> | Best-of-8      | **36.7±0.7** | 36.3±0.6 | 36.5±0.8 |
> | Best-of-16     | **39.1±1.1** | 38.5±0.9 | 38.6±0.6 |
> | Best-of-32     | **40.5±0.8** | 39.8±0.6 | 38.8±0.7 |
> | Best-of-64     | **41.3±0.6** | 40.1±0.8 | 39.7±0.8 |
>
> **Deepseek data on Llama-3.1-8b (corresponding to Figure 3 and Table 6)**
>
> | **GSM8K** |             |         |         |
> |-----------------|------------|---------|---------|
> |                | Ours       | Hard    | Soft    |
> | Best-of-4      | **86.7±0.3** | 86.5±0.4 | 86.3±0.4 |
> | Best-of-8      | **87.3±0.5** | 87.2±0.3 | 87.0±0.4 |
> | Best-of-16     | **88.5±0.5** | 88.0±0.3 | 88.1±0.3 |
> | Best-of-32     | **89.2±0.3** | 88.7±0.6 | 88.9±0.4 |
> | Best-of-64     | **90.0±0.4** | 89.3±0.4 | 89.6±0.5 |
>
> | **MATH500** |             |         |         |
> |-----------------|------------|---------|---------|
> |                | Ours       | Hard    | Soft    |
> | Best-of-4      | **50.8±0.4** | 50.5±0.6 | 50.2±0.9 |
> | Best-of-8      | **52.1±0.5** | 51.8±0.7 | 50.8±0.7 |
> | Best-of-16     | **52.9±0.3** | 52.7±0.3 | 50.5±1.0 |
> | Best-of-32     | **53.1±0.5** | 52.5±0.5 | 51.2±0.9 |
> | Best-of-64     | **53.5±0.5** | 53.1±0.4 | 50.9±0.8 |
>
> **Mistral data on Llama-3.2-1b-Instruct (corresponding to Table 2)**
>
> | **GSM8K** |             |         |         |
> |-----------------|------------|---------|---------|
> |                | Ours       | Hard    | Soft    |
> | Best-of-4      | 82.1±0.2 | 81.9±0.2 | **82.2±0.3** |
> | Best-of-8      | **82.6±0.1** | 82.2±0.4 | 82.3±0.2 |
> | Best-of-16     | **83.5±0.3** | 82.8±0.3 | 83.2±0.2 |
> | Best-of-32     | **84.1±0.3** | 83.8±0.4 | 83.6±0.5 |
> | Best-of-64     | **84.0±0.3** | 83.9±0.4 | 83.8±0.4 |
>
> | **MATH500** |             |         |         |
> |-----------------|------------|---------|---------|
> |                | Ours       | Hard    | Soft    |
> | Best-of-4      | **32.3±0.5** | 31.5±0.5 | 32.0±0.6 |
> | Best-of-8      | **34.2±0.5** | 33.5±0.4 | 33.7±0.5 |
> | Best-of-16     | **37.0±0.6** | 36.3±0.7 | 35.4±0.6 |
> | Best-of-32     | **36.4±0.8** | 36.2±1.1 | 35.9±0.7 |
> | Best-of-64     | **36.3±0.9** | 35.9±1.3 | 35.5±0.9 |
>
>
> **Mistral data on Llama-3.2-3b-Instruct (corresponding to Table 3)**
>
> | **GSM8K** |             |         |         |
> |-----------------|------------|---------|---------|
> |                | Ours       | Hard    | Soft    |
> | Best-of-4      | **83.6±0.3** | 83.3±0.3 | 83.3±0.3 |
> | Best-of-8      | **84.1±0.4** | 83.5±0.4 | 83.7±0.5 |
> | Best-of-16     | **84.5±0.3** | 83.9±0.4 | 84.0±0.2 |
> | Best-of-32     | **85.0±0.2** | 84.4±0.5 | 84.6±0.4 |
> | Best-of-64     | **85.4±0.4** | 84.8±0.3 | 85.0±0.4 |
>
> | **MATH500** |             |         |         |
> |-----------------|------------|---------|---------|
> |                | Ours       | Hard    | Soft    |
> | Best-of-4      | **35.6±0.6** | 35.3±0.7 | 35.2±0.4 |
> | Best-of-8      | **37.7±0.8** | 37.4±0.5 | 37.1±0.8 |
> | Best-of-16     | **38.9±0.6** | 38.3±0.6 | 38.2±0.7 |
> | Best-of-32     | **40.8±1.1** | 40.1±0.8 | 39.6±0.9 |
> | Best-of-64     | **41.9±1.0** | 41.1±0.8 | 40.2±0.8 |
>
> We would like to convey that our primary contribution is the development of a mathematically rigorous framework that directly connects process rewards to KL-regularized MDPs, with experiments demonstrating the effectiveness.
> In the future version of our paper, we would add all the additional experiments with standard deviation to provide clearer evidence of our claim.

---

> ### Author Response · Authors · 2025-03-05
> **Regarding the Weakness of Our Paper**
>
> Dear Reviewer kNgM,
>
> **Regarding Effectiveness on Complex Tasks and Performance Gains**
>
> Our work introduces a theoretically principled approach to process reward modeling through entropy regularization. While we observe consistent performance improvements across our experiments, we recognize that the numerical magnitude of these gains (0.5-2%) merits a more nuanced discussion.
>
> The primary contribution of our work is the development of a mathematically rigorous framework that directly connects process rewards to KL-regularized MDPs. This theoretical foundation offers several key advantages that previous methods lack:
>
> **1. Mathematical Guarantee of Optimality:**
> Our derivation shows that the entropy-regularized process reward directly optimizes the KL-regularized objective, providing theoretical guarantees for the reward formulation.
>
> **2. Dual Formulation Flexibility:**
> The derived dual formulation:
> soft-max when sampling from weak initial policy,
>
> $\pi_0$, $r(a^{L}, x) = \frac{1}{\eta}\ln\mathbb{E}_{a^{-[l]} \sim \pi_0}e^{\eta r(a,x)}$.
>
> soft-min when sampling from strong optimal policy $pi_*$,
>
> $r(a^{L}, x)= -\frac{1}{\eta}\ln\mathbb{E}_{a^{-[l]} \sim \pi^*} e^{-\eta r(a,x)}$.
>
> allows adaptive behavior depending on the policy context, which is a theoretical property not present in previous approaches, facilitating more robust and flexible learning for models with different capabilities.
>
> **3. Enhanced Signal Amplification:**
>
> Traditional methods typically require sampling from an optimal policy model $\pi_*$, but our method can effectively operate with an initial weak policy $\pi_0$ by amplifying sparse rewards. In traditional process reward methods, the reward is computed as $r(a^{-L}) = \mathbb{E}_{\pi_0}[r(a|a^{-L})]$, which can yield weak signals when $\pi_0$​ samples from a small, concentrated region of the solution space.
>
> Drawing from principles similar to importance sampling in reinforcement learning, our method effectively uses the ratio between target and initial policies $\frac{\pi_*}{\pi_0}$. By applying exponential transformation $e^{\eta r(a,x)}$ before expectation and logarithmic transformation after, our approach smooths out reward signals and effectively redistributes reward across the policy space. Therefore, our method also provides independence from optimal policy.
>
> **4. Single Hyperparameter Control:**
> The $\eta$ parameter provides an elegant theoretical mechanism to balance exploitation and exploration, as demonstrated by our finding of different optimal values for datasets of varying difficulty.
>
> The consistent improvements we observe across diverse experimental settings (different models, datasets, and evaluation methodologies) serve primarily to validate these theoretical properties. In our revision, we will:
>
> 1. Refine our discussion of performance gains to more precisely reflect the empirical evidence
> 2. Revise our dictions (i.e. significant) and claims to better align our initial motivation that focuses on the theoretical framework.
> 3. More explicitly connect the empirical results to the theoretical advantages of our approach
>
> We believe our work makes a valuable contribution by providing a theoretically grounded approach to process reward modeling that addresses fundamental limitations in previous methods, with empirical results that consistently validate the theoretical framework.
>
> **Clarification on Training and Evaluation Methodology**
>
> To be absolutely clear: All results reported in our paper, including Table 1, were evaluated on the test sets of GSM8K and MATH, not on training data. Our experimental protocol in section 5.2 was as follows:
>
> We initially attempted to align our policy models (Mistral-MetaMath-7b and deepseek-math-7b-instruct) using only the GSM8K and MATH training sets for RLHF.
> During this process, we observed that while the models showed improvement on the training data, the gains on the test sets were marginal. This pattern is consistent with overfitting, as the models had already been exposed to these datasets during their original supervised fine-tuning.
>
> To address this, we incorporated 12,500 additional novel questions from OpenMathInstruct-2 into our RLHF training data. This dataset contains synthetic mathematical problems and rationales not present in the original training data of our policy models.
> We conducted a blended training approach, mixing this new data with questions from the GSM8K and MATH training sets to prevent catastrophic forgetting.
>
> The performance of all models (baseline and RLHF-enhanced) was then evaluated exclusively on the test sets of GSM8K and MATH.
> The OpenMathInstruct-2 data was used only for training, not for evaluation. The test sets remained completely unchanged throughout our experiments. We will revise our manuscript to make this methodology clearer.

---

> > ### Comment · Reviewer_kNgM · 2025-03-06
> > **reply to authors**
> >
> > Dear authors,
> >
> > Thank you very much for running these additional experiments, it is reassuring to see your methods still performs better than other baselines, though the difference is not drastic, it is still a small win. I encourage you to revise the paper with these additional results and to (as you correctly identified): "_Refine your discussion of performance gains to more precisely reflect the empirical evidence_"
> >
> > Furthermore, thank you for clarifying the theoretical advantages of your approach, these motivations were not initially clear in the paper.
> >
> > Thanks also for clarifying the train / test experimental design, and sorry for my confusion. What you did is sound and a good idea. Please make sure to clarify this also in the paper.
> >
> > I will update my rating and ask authors to update their manuscript accordingly (like they said they will):
> > - Revise dictions (i.e. significant) and claims to better align their initial motivation that focuses on the theoretical framework.
> > - More explicitly connect the empirical results to the theoretical advantages of their approach

---

### Review · Reviewer_74eA · 2025-03-09

**Summary Of Contributions:**

The paper introduces an Entropy-Regularized Process Reward Model (ER-PRM) for improving mathematical reasoning in LLMs. It builds on process reward models (PRMs), which assign rewards to each reasoning step rather than just the final answer. The core idea is to use KL-regularized Markov Decision Processes (MDPs) to ensure that the model does not deviate too much from its initial policy while optimizing for reasoning correctness. The reward formulation uses entropy regularization, balancing reward maximization with policy stability.

The paper evaluates two generator LLMs viz., Mistral-7B fine-tuned on MetaMath dataset and DeepSeek-Math-7B-Instruct. The reward models used are Llama-3.2-3B and Llama-3.2-1B. The results demonstrate superior reasoning accuracy over baseline reward models like hard-label PRM, soft-label PRM and outcome reward model (ORM), on the MATH and GSM8K datasets. The authors also use ER-PRM in RLHF training, yielding further gains in reasoning accuracy. The models are evaluated on MATH and GSM8K datasets, w.r.t. the best-of-n and top-1 accuracy. Overall, the proposed method achieves 1% improvement on GSM8K and 2-3% on MATH under best-of-N evaluation. It is observed that ER-PRM consistently outperforms these baselines, especially for complex reasoning tasks. This approach strengthens LLMs' stepwise reasoning capabilities by ensuring a stable optimization process and providing fine-grained step-level supervision using entropy-regularized reward functions.

**Audience:**

Yes

**Claims And Evidence:**

Yes

**Requested Changes:**

Refer weaknesses.

The paper introduces an entropy-regularized process reward model (ER-PRM) with a solid theoretical foundation and some improvements in mathematical reasoning tasks, though the gains are small. However, it lacks originality as it primarily builds on existing methods with KL-regularized MDPs. Additionally, the paper needs stronger comparisons with top-performing models, clearer organization in the experimental and results sections, and a better justification for the choice of models. These issues should be addressed in the paper.

**Strengths And Weaknesses:**

Strengths:
--------------------
- The paper introduces an entropy-regularized process reward model (ER-PRM) that builds on existing process reward models by incorporating KL-regularized MDPs. This approach effectively balances reward optimization with policy stability, ensuring that the model does not deviate excessively from its initial distribution. The theoretical grounding and mathematical derivations provide a solid foundation for this method.

- The proposed method is evaluated on two widely used mathematical reasoning benchmarks, MATH and GSM8K, demonstrating consistent improvements over baseline process reward models. The results indicate that ER-PRM enhances reasoning performance, particularly on complex mathematical problems, with 1% improvement on GSM8K and 2-3% improvement on MATH under best-of-N evaluation. The additional experiments with RLHF fine-tuning further highlight the model’s ability to improve reasoning accuracy in LLMs.

Weaknesses or Concerns:
--------------------
- The proposed ER-PRM method lacks significant contribution and novelty. It is essentially a variant of the soft-label PRM, incorporating KL-regularized MDPs to prevent the model from deviating excessively from its initial policy during optimization. Furthermore, the percentage improvement on the MATH and GSM8K datasets is not substantial.
The paper should also compare the results of the proposed method with those of existing state-of-the-art approaches on MATH and GSM8K. Additionally, providing a comparison with current state-of-the-art methods and highlighting how close the proposed results are to them would offer valuable insights for readers.

- Sections 4 and 5 require substantial improvement in terms of structure and clarity. The authors should create distinct subsections (or use paragraph headings) to separately discuss the different generator models and reward models used. There are inconsistencies regarding which LLM was used as the reward model (see the point below). Additionally, Section 4 does not justify the choice of LLMs. Without a clear justification, it remains unclear whether the proposed ER-PRM method is generalizable to other LLMs.

- The paper contains grammatical errors, such as in the sentence: "When the models are trained on deepseek data and evaluate on deepseek-math-7b-instruct..." Additionally, some sentences are poorly constructed, such as the first two sentences in the second paragraph of page 7.

- The paper states: "The reward models trained on data from Mistral and deepseek are denoted as..." In Section 5, the authors repeatedly refer to "deepseek data." It is recommended to cite appropriate sources, as "deepseek data" is a vague term given DeepSeek's recent release of multiple models and datasets.

- In Section 4, the authors state: "We train our reward models using Llama-3.1-8B." However, Table 2 and Table 3 present results for reward models trained on Llama-3.2-1B-Instruct and Llama-3.2-3B-Instruct, respectively. This discrepancy should be clarified.

- In Section 2, the authors mention: "However, PRMs require extensive human annotation, which is challenging for open-source efforts." In this paper, the authors use an LLM as the reward generator. However, how reliable is this reward modeling? Since the reward generator is not inherently sound, it may assign low rewards even to high-quality responses. This also raises concerns about generalizability, as noted earlier.

---

> ### Author Response · Authors · 2025-03-12
> **Response to Reviewer 74eA Regarding the Writing and Organization**
>
> Dear Reviewer 74eA,
>
> Thank you for your review of our paper. We appreciate your recognition and your suggestions. Below, we address your concerns and suggestions in detail.
>
> **Sections 4 and 5: Structure and Clarity**
>
> We sincerely appreciate the reviewer’s thoughtful comments regarding the organization of Sections 4 and 5. We have made a series of revisions aimed at enhancing the clarity and readability of these sections.
>
> Firstly, we have added headings to most of the paragraphs, each of which reflects the focus of that particular section. This will help readers more easily navigate through the content and quickly understand the main concepts being discussed. We believe them improve the overall flow and organization.
>
> Secondly, to provide a clearer introduction to the data, models, and baselines in our work, we have incorporated multiple bulleted lists. These lists serve to outline the key components, making it easier for readers to grasp the essential details.
>
> Thirdly, we have carefully restructured sentences to eliminate grammatical errors that may have been present in the original version. Additionally, we have removed overly lengthy sentences, which we believe will significantly improve the readability and coherence of the sections. We have focused on making the text more concise while retaining the necessary technical depth and ensuring that our argument is communicated effectively.
>
> We believe these changes collectively make the revised sections more accessible and address the reviewer's concerns. We hope the updated version meets the reviewer’s expectations and provides a clearer presentation of our work.
>
> **Grammatical Errors and Sentence Construction**
>
> Thank you for pointing out the grammatical errors and sentence constructions. We have carefully reviewed every sentence in the paper and addressed the identified issues. In the newly uploaded version, we have made substantial improvements to enhance clarity and readability, particularly in Sections 4 and 5. We sincerely appreciate your time and feedback, and we hope you find the revised version clearer and more polished. We would greatly appreciate it if you could take another look at the updated manuscript.
>
> **Proper Citation of the term "deepseek data"**
>
> We appreciate the reviewer’s concern regarding the vague reference to "deepseek data." In the revised manuscript, we have provided a clear definition of both deepseek-data and mistral-data:
> - Mistral-data: We use Mistral-MetaMath-7b as the generator and completer to collect the process reward training data.
> - deepseek-data: We use deepseek-math-7b-instruct as the generator and completer to collect the process reward training data.
>
> We believe this clarification of the specific datasets used and the citation of appropriate sources will ensure clarity and accuracy.
>
> **Discrepancy between Section 4 and Tables 2 and 3**
>
> Thanks for pointing that out. In Section 4, we mention that we would use Llama-3.1-8B to train the reward models, which corresponds to our main experiments in Section 5.1. The experiment results are shown in Figures 2 and 3. We also list the results in Tables 5 and 6 in the Appendix.
>
> In the newly uploaded version, we have modified this part to provide clearer guidance for the reader of every experiment result. Specifically:
>
> In Section 4, we add:
>
> We train the reward models using Llama-3.1-8B on Mistral-data and deepseek-data in our main experiments (Section 5). In Section 6.1, we explore the reward model training with various sizes using Llama-3.2-1B-Instruct and Llama-3.2-3B-Instruct on Mistral-data.
>
> In Section 5, we clearly state that:
>
> We train the reward model with Llama-3.1-8B as the base model. The results of our model compared with several baseline methods are shown in Figure 2 and 3, which are trained in an auto-regressive way. The details are also listed in Table 5 and 6 in the Appendix.
>
> **Reliability of reward modeling and generalizability**
>
> Thank you for raising this point. We would like to clarify that in Section 2, we present the related work in a chronological manner. Initially, researchers relied solely on human annotation for labeling step rewards, which is why we mention "However, PRMs require extensive human annotation, which is challenging for open-source efforts." However, subsequent studies have shown that automatic labeling can achieve comparable or even superior data quality. For example, in Math-Shepherd (Wang et al., 2024), experiments were conducted comparing LLM-labeled data with human-labeled data, showing that LLM-labeled data could be used to train a more effective reward model. Our work primarily builds on this popular automatic labeling framework, and we believe the quality of our data is at least on par with human-labeled data.
>
> We have also modified a bit of Section 2, to introduce the feasibility of the automatic labeling method.
>
> [1] Wang et al., 2024, Math-Shepherd: Verify and Reinforce LLMs Step-by-step without Human Annotations

---

> ### Author Response · Authors · 2025-03-13
> **Response to Reviewer 74eA Regarding the Novelty and Contribution**
>
> Dear Reviewer 74eA,
>
> We would like to thank you for the thoughtful assessment of our work. Below, we explain the contribution and the novelty of our work.
>
> The contribution and the novelty of our work lie in the theoretical framework of the derivation of process rewards through entropy regularization within the KL-regularized MDP framework, instead of solely improving the numerical performance. This theoretical grounding provides several distinct advantages that explain our method's improved performance:
>
> **Dual Formulation: Soft-Max and Soft-Min**
>
> A key theoretical strength of our approach is the flexibility of a dual formulation. When sampling completions from the initial policy $\pi_0$, our process reward implements soft-max:
>
> $r(a^{L}, x) = \frac{1}{\eta}\ln\mathbb{E}_{a^{-[l]} \sim \pi_0}e^{\eta r(a,x)}$.
>
> Conversely, when sampling from the optimal policy $\pi_*$, it implements soft-min:
>
> $r(a^{L}, x)= -\frac{1}{\eta}\ln\mathbb{E}_{a^{-[l]} \sim \pi^*} e^{-\eta r(a,x)}$.
>
> This duality provides theoretical flexibility not present in previous methods. It allows our approach to adapt optimally depending on the policy being used for sampling, facilitating more robust learning across a spectrum of model capabilities.
>
> **Signal Amplification for Smoothing Sparse Rewards**
>
> The previous methods normally require sampling from optimal policy model $\pi_*$, but our method can rely on initial weak policy $\pi_0$ by amplifying the sparse rewards with our ER-PRM. In traditional process reward methods, the reward $r(a^{−L})$ is computed as an expectation under the initial policy $\pi_0$ , such that: $r(a^{-L}) = \mathbb{E}_{\pi_0}[r(a|a^{-L})]$.
>
> While this approach is straightforward, the sole reliance on the initial policy $\pi_0$ can result in sharp or overly localized reward distributions. This limitation is particularly evident when $\pi_0$ samples from a small, concentrated region, leading to weak or sparse reward signals.
>
> Drawing from similar experience with the importance sampling in reinforcement learning, the reward can be estimated as a ratio between the target policy $\pi_*$ and the initial policy $\pi_0$, specifically: $r = \log{\frac{\pi_*}{\pi_0}}$.
>
> In our work, we adopt this definition and extend it with exponential transformation $e^{\eta r(a,x)}$ before expectation and logarithmic transformation afterward. This approach effectively smooths out the reward signal, addressing the shortcomings of traditional process reward models. For instance, when $\pi_0$ reasons over complex mathematical tasks where correct solutions form a small subset of the possible answer space, this amplification is theoretically crucial for effective learning. Our formulation amplifies these weak signals through the ratio $\frac{\pi*}{\pi_0}$, redistributing the reward more effectively across the policy space. This smoothing effect allows for a more robust reward calculation, leading to stronger step-level signals and better optimization during training.
>
> **Simplicity in Hyperparameter and Independence of Optimal Policy**
>
> Our method only requires turning a single hyperparameter $\eta$ that controls the entropy penalty. This provides a theoretically elegant way to balance between:
> 1. Exploitation of high-reward trajectories (higher $\eta$ values)
> 2. Exploration across the policy space (lower $\eta$ values)
>
> Another significant theoretical advantage of our method is that, unlike traditional approaches that depend on sampling from the optimal policy $\pi_*$​, our entropy-regularized process reward can be computed using only the initial policy $\pi_0$​. This removes the dependency problem presented.
>
> We believe these theoretical foundations provide a solid theoretical framework for the popular PRM method nowadays.

---

### Decision · Action_Editor_cwrR · 2025-05-04

**Recommendation:** Accept with minor revision

**Comment:**

The submission received three detailed reviews, all of which asked for clarifications and highlighted strengths and weaknesses of the paper. The authors responded in detail to those and updated the paper accordingly. All three reviewers finally recommended the acceptance of the submission. Nevertheless, two of the reviewers still remarked on the limited novelty and significance of the submission which is, however, not a central decision criteria.

Hence, I am suggesting acceptance of the paper with minor revisions, which are detailed below and are based on the reviewers' comments but also my own reading of the submitted paper:
* Although grammar of the manuscript improved a lot already, another careful update of the paper in that regard should be conducted.
* Update section 3.2 to be easier comprehensible and have a clear logical structure (e.g., currently the authors start by saying "we define the reward by ..." just to say we obtain "..." (essentially the assumed reward function). Later they say that any policy can be computed using any policy and state an equation for the reward. Please show how you arrive at this equation. Also clarify, and this is important as you extensively remarked on this in the discussion with the authors, how the difference in the equations for the reference policy and optimal policy come about. Also explain what happens in the mentioned case that "More generally, the reward can be computed using any policy including the optimal policy" or adjust this statement.
* Clarify the computation of the confidence intervals for the best-of-n approach.
* Include confidence intervals in all results tables (in particular, also in Tables 1 and 4), and, if possible, also in the figures.
* The clarity of the experimental setting could be improved by showing a chart or similar depicting how all components and models come together.

**Audience:**

Yes, improving the performance of process reward models for LLM reasoning can help to better understand these models and to develop better models more effectively.

**Claims And Evidence:**

The claims are, after some updates during the review and discussion phase, largely sufficiently supported by evidence. One point that still needs to picked up by the authors is to include confidence interval information into how they talk about their results - not all improvements are significant.